# Epithelial TGFβ engages growth-factor signalling to circumvent apoptosis and drive intestinal tumourigenesis with aggressive features

The pro-tumourigenic role of epithelial TGFβ signalling in colorectal cancer (CRC) is controversial. Here, we identify a cohort of *born to be bad* early-stage (T1) colorectal tumours, with aggressive features and a propensity to disseminate early, that are characterised by high epithelial cell-intrinsic TGFβ signalling. In the presence of concurrent *Apc* and *Kras* mutations, activation of epithelial TGFβ signalling rampantly accelerates tumourigenesis and share transcriptional signatures with those of the *born to be bad* T1 human tumours and predicts recurrence in stage II CRC. Mechanistically, epithelial TGFβ signalling induces a growth-promoting EGFR-signalling module that synergises with mutant APC and KRAS to drive MAPK signalling that re-sensitise tumour cells to MEK and/or EGFR inhibitors. Together, we identify epithelial TGFβ signalling both as a determinant of early dissemination and a potential therapeutic vulnerability of CRC's with *born to be bad* traits.

In 2020, there were ~1.9 million newly diagnosed cases of colorectal cancer (CRC) worldwide[1], with recent projections estimated that the annual global burden will exceed 2.2 million new cases by 2030[2]. The recent widespread implementation of population-based CRC screening programmes has increased the detection of asymptomatic adenomas and early-stage CRC, offering a tangible opportunity to intervene early, improve prognosis and quality of life, and reduce mortality. An important corollary of systematic screening programmes is that the incidence of stage-I CRC [defined as tumours that have grown into the submucosa (T1) or the muscularis propria (T2), but have not spread to regional lymph node (N0) or distant organs (M0)] has increased nearly four-fold, approximating 42–45% of all new CRC diagnoses[3]. Previously, such asymptomatic cases were often only diagnosed at the metastatic stage, where 5-year survival remains dismal (<10%)[4]. Histological assessment of resected T1 tumour tissue is unable to reliably identify aggressive lesions; therefore, a sizeable subset of patients have an unidentified and unquantified risk of metastatic spread. The ability to stratify endoscopic resection patients, according to their risk of disease relapse, has the potential to spare low-risk patients from the potentially toxic side-effects of unwarranted therapy and to guide follow-up surveillance colonoscopies and treatment decisions for high-risk early-stage CRC patients. These considerations underscore an urgent need for the validation of transcriptional biomarkers that enable the prospective diagnosis, biological characterisation and pre-clinical modelling of poor-prognosis early-stage tumours, and the identification of druggable therapeutic targets to prevent the progression or relapse of early-stage disease.

Whereas early detection can markedly improve the chances of survival, a subset of T1 CRCs bestowed with inherent metastatic potential with metastasis-founder cells disseminating even before the primary tumour is clinically detectable[5]. The existence of these so-called *born to be bad* CRCs, has shifted the paradigm away from sequential metastatic cascades to the early dissemination of metastasis-competent primary tumour cells[6,7]. Leveraging transcriptional profiling of a unique T1 CRC cohort to guide the development of an autochthonous mouse model of early-stage, dissemination-prone CRC, we sought to understand and target the early genetic aberrations

✉e-mail: dustin.flanagan@monash.edu; o.sansom@beatson.gla.ac.uk

and pathway dysregulation that underpin an aggressive disease course.

The development of CRC is underpinned by the accumulation of loss- or gain-of-function mutations in key driver genes (e.g. *APC*, *KRAS*, *TP53* and *SMAD4*), which collectively disrupt signalling pathways that impinge on the proliferation and survival of differentiated and stem/progenitor cells within the intestinal epithelium[8]. Among the earliest genetic events in adenoma formation, mutations in the *APC* tumour suppressor and the *KRAS* proto-oncogene cause hyperactivation of the pro-proliferative Wnt and MAPK signalling pathways, respectively. In addition, TGFβ signalling plays complex, context- and stage-dependent roles in the pathogenesis of CRC, functioning as a tumour suppressor in premalignant colonic epithelial cells but switching to a tumour promoter in more advanced stages of the disease[9].

TGFβ signalling is initiated upon binding of secreted TGFβ ligands to the type II serine/threonine kinase receptor TGFBR2, which subsequently recruits and phosphorylates the type I receptor TGFBR1 (also known and referred to herein as ALK5). This in turn leads to the activation of canonical SMAD-dependent and non-canonical SMAD-independent downstream pathways, eliciting a multitude of cellular responses including growth arrest, apoptosis, and differentiation[9]. Inactivating mutations in genes encoding components of the TGFβ signalling pathway feature as common events in CRC and can occur both at the cell-receptor level (*TGFBR2* frameshift mutations in microsatellite-unstable/mismatch repair-deficient CRC) and the transcriptional-effector level (mutations in *SMAD4*—the common denominator of canonical TGFβ signalling—in microsatellite stable CRC)[10,11]. Numerous studies, using genetically-engineered-mouse models (GEMMs), have shown loss of *Smad4* or *Tgfbr1/2*, in concert with other key mutational drivers of CRC (loss of *Apc* and/or expression of oncogenic KRAS), accelerates intestinal tumourigenesis and promotes tumour progression, positioning TGFβ signalling as a fundamental tumour suppressor in CRC[12–14].

On the other hand, the aberrant activation of TGFβ within the tumour microenvironment (TME) exerts pleiotropic pro-tumourigenic effects. Indeed, it is hypothesised that the selective loss of TGFβ-mediated cytostatic responsiveness in the tumour epithelium allows for increased ligand production and secretion from the TME, which in turn promotes extracellular matrix remodelling, activation of cancer-associated fibroblasts (CAFs), epithelial-mesenchymal transition (EMT), angiogenesis, immune evasion, metastatic competence and drug resistance[15–17]. Importantly, these disparate processes go some way to explain how tumour cells evade TGFβ-mediated tumour-suppression and co-opt pro-oncogenic TGFβ functions to further tumour progression. Nonetheless, the essential role for TGFβ in driving tumour progression is manifested in its capacity as a key determinant of the most aggressive molecular subtype of CRC, consensus molecular subtype 4 (CMS4), which is characterised by pro-tumourigenic stromal TGFβ signalling and aberrant EMT-pathway activation[18].

In parallel to elegant studies underscoring the dichotomous role of TGFβ signalling in CRC[19], recent data demonstrate that canonical TGFβ signalling promotes colonic epithelial repair post damage[20] suggesting that even during homoeostasis and response to injury, TGFβ may not always act as an epithelial tumour suppressor. Consistent with this notion, the derivation of CRC cell-intrinsic subtypes (CRIS) reveals that poor-prognosis patients characterised by epithelial cell-autonomous TGFβ signalling (CRIS-B)[21], although no associations with mutations in *SMAD* or *TGFBR* genes were noted. Moreover, epithelial tumour cell-intrinsic activation of TGFβ signalling is associated with poor prognosis and therapy resistance[15], and is required to drive human *BRAF*-mutant serrated adenoma organoids towards the metastatic CMS4 subtype[22].

We have previously activated TGFβ signalling within the epithelium of different tissues using a conditional active *Tgfbr1/Alk5* allele (*Hprt*-LSL-TβR1^CA), resulting in diverse outcomes. Targeted activation of this allele in the liver or pancreas resulted in a dramatic perturbation of homoeostasis, the induction of p21 and other senescence markers and damage to both tissues, consistent with the negative regulation of growth/survival pathways[23,24]. Interestingly, in the pancreas, coupling of active TGFβ signalling with oncogenic KRAS promoted acinar to ductal metaplasia, alongside the pronounced perturbation of tissue homoeostasis. This indicates in specific contexts, active TGFβ signalling may also function as a tumour-promoter in the epithelium[23]. Intriguingly, activation of this same allele within the ovarian stroma led to the development of ovarian stromal tumours[25]. These findings are consistent with differential roles for TGFβ signalling in epithelial versus stromal tissues.

In this study, we molecularly characterise tumours from a cohort of T1 CRC patients enriched for *born to be bad* lesions, and identify that aggressive traits are associated with elevated epithelial TGFβ signalling rather than stromal biology. We demonstrate a potent synergy between common CRC mutations in *Apc* and *Kras* (together, though not alone) and epithelial-specific activation of TGFβ signalling in promoting intestinal tumourigenesis. Critically, the resulting tumours overlap transcriptionally with early-disseminating human T1 lesions and poor-prognosis stage II CRCs, lending support for the translation potential of epithelial TGFβ as an actionable, predictive biomarker in the early-stage setting and for extrapolating the efficacious therapeutic approaches, achieved in our models, to the clinic.

## Results

### Early disseminating T1 patient cohort

Utilising the Northern Ireland Cancer Registry (NICR) database, we identified patients ($n = 10$), diagnosed with T1 CRC (SNOMED codes C18-C20) between 2007-2012, who subsequently relapsed or died from CRC (defined hereafter as 'relapse' cases). Most of these patients had undergone radical surgery rather than local excision. Using a nested case-control methodology, CRC deaths/recurrences within the T1 cohort were matched by gender, 10-year age-band, and diagnostic hospital lab to control patients ($n = 2–3$), who had also been diagnosed with T1 CRC but who did not experience disease relapse or CRC-related death (defined hereafter as 'non-relapse' cases). Following pathological verification of T1 status (pT1), alongside exclusions due to limited availability of tumour tissue and extracted RNA/DNA, transcriptional profiles were generated for 27 patients (10 relapse, 17 non-relapse; Supplementary Table 1) and mutational profiles were determined for 21 patients (8 relapse, 13 non-relapse, Supplementary Fig. 1a).

### Epithelial TGFβ signalling, rather than stromal histology or molecular subtype, is associated with relapse in early disseminating T1 CRC

Following data processing and normalisation (detailed in Methods), unsupervised PCA assessment of all transcripts in primary tumour samples ($n = 27$) revealed a qualitative separation between transcriptional profiles from relapse and non-relapse tumours (Supplementary Fig. 1b). Given that stromal and immune infiltration holds prognostic value in stage II/III CRC, we used the Microenvironment Cell Populations-(MCP)-counter algorithm to quantify the abundance of fibroblasts and cytotoxic T lymphocytes in the TME of pT1 tumours. However, we found no significant association between the MCP-counter scores for these two populations and relapse status (Fig. 1a). To enumerate the stromal cell content in situ, we further analysed haematoxylin and eosin-(H&E)-stained whole slide images of our pT1 samples using the open-source quantitative pathology software QuPath[26]. These analyses revealed a significant positive correlation between transcriptome-based stromal (fibroblast) MCP-counter scores and the digital pathology-based stromal scores (Fig. 1b, rho = 0.69, $p = 0.00009$), validating our in silico approach to quantifying stromal cell abundance in the TME. Together, these findings argued against an early prognostic value for stromal content in our pT1 cohort

and prompted us to investigate tumour-intrinsic factors that may contribute to relapse.

As the development of CRC entails the accumulation of mutations in key driver genes, which often correlate with distinct stages of tumour development[27], we next compared the mutational profiles of relapse and non-relapse cases in our pT1 patient cohort. These analyses revealed a lower incidence of *TP53* mutations in relapse cases (4/8; 50%), compared with non-relapse (11/13; 85%), alongside an enrichment of *KRAS* mutations in relapse cases (5/8; 63%), compared with non-relapse (4/13; 31%). However, less than 80% of the samples had matching mutational profiles due to the limited amounts of tissue available for sequencing (Fig. 1c), hampering our ability to draw conclusions across the entire cohort.

To identify differentially enriched biological pathways between relapse and non-relapse pT1 cases, we used gene set enrichment analysis (GSEA) to interrogate CRC-associated Hallmark and curated gene sets[22,28]. In line with the observed mutational profiles, we found that relapse cases were significantly enriched for genes involved in KRAS and mTORC1 signalling (Fig. 1d). We next applied the CMS and CRIS transcriptomic classifiers to our pT1 cohort, with the caveat that both CMS and CRIS molecular subtyping can suffer from poor statistical confidence in small cohorts such as ours. Although both of these classification algorithms are effective in stratifying later-stage disease, linking both CMS4 and CRIS-B to poor prognosis[18,21], neither of these two classifiers was able to identify pT1 relapse cases (Supplementary Fig. 1c). To account for the small sample sizes, we utilised the CRIS and CMS classifiers in conjunction with GSEA. From these analyses, we observed uniformly elevated CRIS signalling across all relapse cases, compared with matched non-relapse samples, suggesting that aggressive pT1 tumours have activated multiple tumour-intrinsic oncogenic signalling pathways (Fig. 1e and Supplementary Fig. 1d, $p_{adj}$ = 5.6e-09; 0.3; 0.1; 0.12 and 0.003, respectively). By applying the CMS classifier, we found that, in line with our stromal assessments (Fig. 1a), aggressive pT1 tumours have significant activation of epithelial cell signalling pathways (CMS2, $p_{adj}$ = 0.01 and CMS3, $p_{adj}$ = 2.6e-09) and, strikingly, a significant suppression of stromal cell signalling that is typical of the mesenchymal/fibroblast-like CMS4 subtype (Supplementary Fig. 1d, CMS1, $p_{adj}$ = 0.003 and CMS4, $p_{adj}$ = 3.7e-10). Similarly, recent transcriptomic profiling studies found that the CMS4 signature is very infrequently observed in early-stage disease, with most premalignant adenomatous polyps and adenomas classified as CMS2 and CMS3, respectively[29,30].

Importantly, we observed enrichment of TGFβ signalling in relapsing pT1 tumours, using the 'Hallmark' TGFβ signalling gene set and a previously defined epithelial-derived 'TGFB_UP' signature[22,28] (Fig. 1f). We and others have previously reported a significant association between TGFβ signalling and the levels of both tumour-infiltrating fibroblasts and cytotoxic immune cells in later-stage disease;[16,21,31] however, we did not observe such an association in our early-stage pT1 tumours (Fig. 1g, rho = −0.1, $p$ = 0.6 and Supplementary Fig. 1e, $r$ = 0.58, $p$ < 2.2e-16). Furthermore, we observed that 'TGFβ-high' relapse-prone pT1 tumours were negatively enriched for gene sets related to CAFs and tumour stroma (Fig. 1h), suggesting that TGFβ signalling is associated with the epithelium (intrinsic, i.e. tumour cell autonomous) rather than the stroma (extrinsic) at early stages of tumour formation. Taken together, our data suggest that activation of epithelial-intrinsic TGFβ signalling is associated with aggressive biology in early-stage pT1 tumours.

## Epithelial cell-intrinsic TGFβ/ALK5 signalling induces apoptosis in vitro but proliferation in vivo

To model the effects of TGFβ-ligand-mediated receptor activation within the intestinal epithelium, we crossed *VillinCre^{ER}* mice[32] with mice carrying a Cre-inducible (*Hprt*-LSL-TβR1^{CA}) transgene knocked into the X chromosome-linked *Hprt* locus[33]. Cre-mediated excision of the floxed STOP (LSL) cassette, the resulting mice (hereafter described as *VilCre^{ER};Alk5*^{CA}) conditionally express a constitutively active form of the TGFβ type I receptor ALK5 (ALK5^{CA}). Importantly, this constitutively active ALK5^{CA} mutant protein remains sensitive to kinase inhibition[33].

To assess the impact of epithelial-intrinsic TGFβ activation, we generated small intestinal organoids from uninduced *VilCre^{ER};Alk5*^{CA} and *VilCre^{ER};Alk5*^{+/+} mice and treated them with 4-hydroxytamoxifen (Tmx) or TGFβ1, respectively. Compared with vehicle-treated controls (EtOH), Tmx-treated *VilCre^{ER};Alk5*^{CA} organoids displayed acute organoid atrophy, reduced viability and increased cell death within 3 days, which was rescued by treatment with the selective ALK5-kinase inhibitor AZ12601011 (ALK5i; Fig. 2a, b and Supplementary Fig. 2a). Of note, whereas ALK5i displays some inhibitory activity against the ALK5-related receptors ALK4 and ALK7, it does not impact BMPR signal transduction[34]. Additionally, Tmx-treated *VilCre^{ER};Alk5*^{CA} organoids increased the expression of TGFβ-target genes compared with vehicle-treated controls (EtOH; Fig. 2c). To model ligand-mediated ALK5 receptor activation in vitro, we treated *VilCre^{ER};Alk5*^{+/+} organoids with recombinant TGFβ1, which induced the expression of bona fide TGFβ-target genes (Fig. 2c) and triggered rapid cell death that was blocked by ALK5i treatment (Fig. 2a, b and Supplementary Fig. 2a). Together, these data demonstrate that conditional expression of the *Alk5*^{CA} allele activates cell-autonomous TGFβ signalling in organoids, faithfully recapitulating the cellular and molecular events following ligand-mediated ALK5 receptor activation in intestinal epithelial cells.

Given that the activation of TGFβ signalling has been implicated in colonic wound-healing post injury[20], we next examined if epithelial TGFβ activation would be better tolerated in vivo than in vitro. For this, we examined the effects of constitutive ALK5-dependent TGFβ signalling on intestinal homoeostasis in *VilCre^{ER};Alk5*^{CA} mice 4 days post-tamoxifen induction. We first evaluated the levels of phosphorylated SMAD3 (p-SMAD3), a key downstream effector of ALK5-dependent TGFβ signalling, and the expression of the TGFβ-responsive genes *Itgβ6* and *Smad7*. Compared to *VilCre^{ER};Alk5*^{+/+} mice, the intestinal epithelium of tamoxifen-treated *VilCre^{ER};Alk5*^{CA} mice was heavily populated with positively stained cells for p-SMAD3, *Itgβ6*- and *Smad7*, consistent with robust activation of TGFβ signalling (Fig. 2d and Supplementary Fig. 2b), whereas gross intestinal morphology and homoeostasis were seemingly unperturbed. To examine more subtle perturbations, we quantified levels of apoptosis, proliferation and differentiation. We observed an increased number of cells staining positive for cleaved caspase-3^{+} in *VilCre^{ER};Alk5*^{CA} mice, compared with *VilCre^{ER};Alk5*^{+/+} mice (CC3^{+} cells; Fig. 2d, e), consistent with a moderate induction of epithelial cell apoptosis in response to constitutive TGFβ/ALK5 signalling. In parallel, *VilCre^{ER};Alk5*^{CA} mice exhibited a marked increase in crypt cell proliferation and the expression domain of *Olfm4*, a robust marker of murine small intestinal stem cells (ISCs)[35,36], suggesting an expansion of the ISC compartment beyond the crypt base (Fig. 2d, f, g). Paneth cells were reduced in number and mislocalized to the villi in *VilCre^{ER};Alk5*^{CA} mice, consistent with the requirement of TGFβ signalling for Paneth cell differentiation[37] (Fig. 2d, h). Interestingly, changes to the stem cell niche (ISC and Paneth cells) were not maintained in *VilCre^{ER};Alk5*^{CA} mice harvested 60 days post-tamoxifen induction (Supplementary Fig. 2c), despite maintaining constitutive TGFβ/ALK5 signalling (data not shown).

To functionally test whether the activation of ALK5-dependent TGFβ signalling was driving these phenotypes through the canonical SMAD4-dependent pathway, we interbred *VilCre^{ER};Alk5*^{CA} mice with *Smad4*^{fl/fl} mice[38] to generate *VilCre^{ER};Alk5*^{CA};*Smad4*^{fl/fl} offspring. Concurrent loss of *Smad4* prevented the phenotypes associated with constitutively active ALK5-dependent TGFβ signalling, restoring apoptosis and crypt cell proliferation to basal levels, rescuing the stem cell expansion and Paneth cell mislocalization, and diminishing TGFβ/SMAD target-gene expression in tamoxifen-treated *VilCre^{ER};Alk5*^{CA};*Smad4*^{fl/fl} mice (Fig. 2e–h and Supplementary Fig. 2b). These

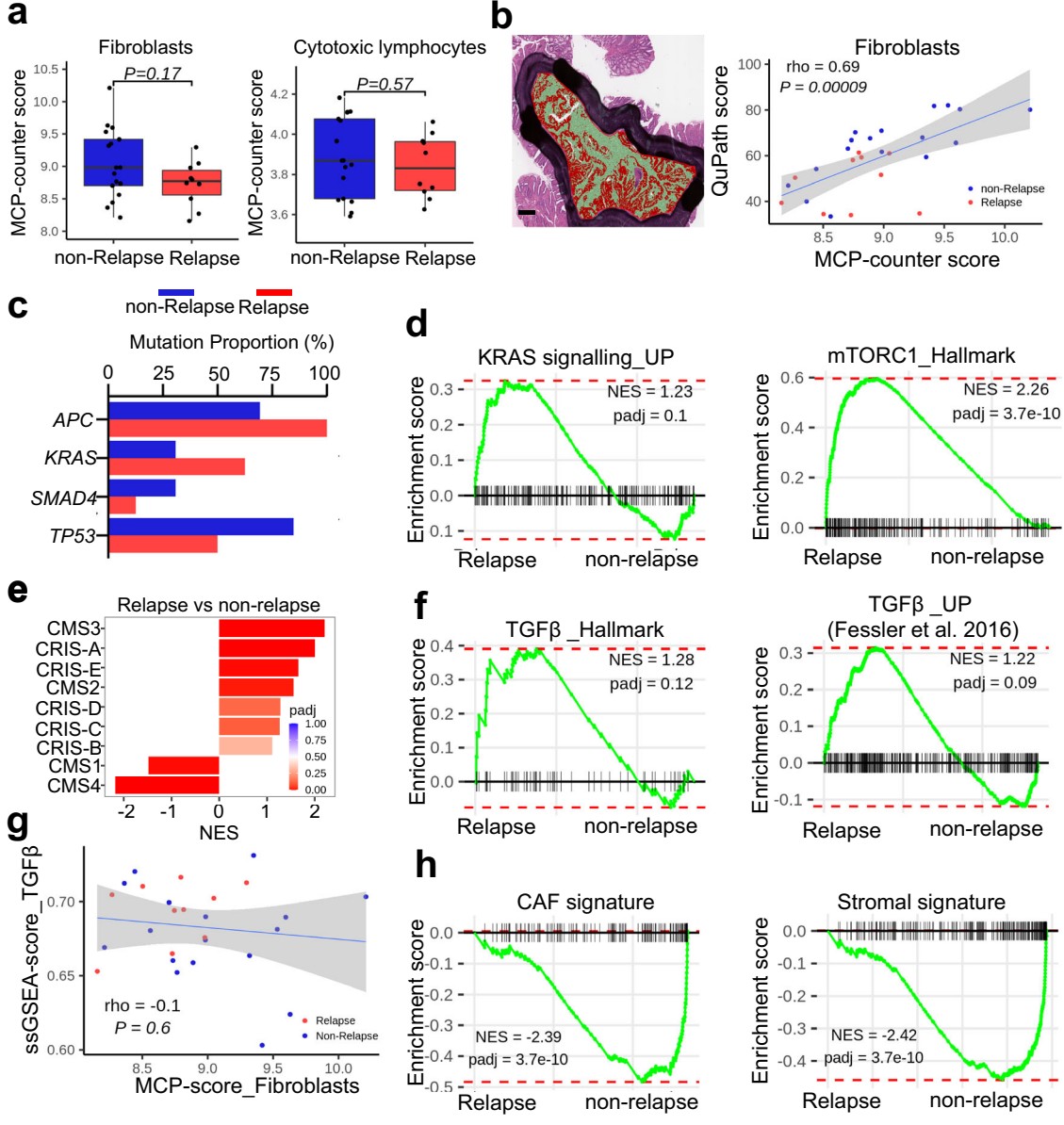

**Fig. 1 | Stromal content has no prognostic value in pT1 CRC. a** Comparison of transcriptome-based Microenvironment Cell Populations (MCP) scores for fibroblasts (left) and cytotoxic T lymphocytes (right) in relapse ($n=10$; red) and non-relapse ($n=17$; blue) pT1 cohort. Statistical significance was determined by two-sided wilcoxon rank-sum exact test using wilcox.test function in stats R package. Horizontal line represents median values, boxes indicate the inter-quartile range and bars denote the maximum and minimum values. **b** Left, representative H&E of pT1 tumour scored for stromal content (red) using QuPath software. Right, correlation between transcriptome-based fibroblast scores and digital pathology stromal scores. Two-sided $P = 0.00009$, rho = Spearman correlation coefficient. Grey region shows the 95% confidence interval (CI) for predictions from a linear model (lm). Scale bar, 250 μm. **c** Proportion of the main mutations in CRCs in relapse (red) and non-relapse (blue) pT1 cohort. **d** Gene set enrichment analysis (GSEA) showing positive enrichment of Hallmark curated gene sets for KRAS and mTORC1 signalling in relapse cases as compared to non-relapse using fast gene set enrichment analysis (fgsea) method. **e** Group-wise GSEA performed using fgsea R

package on the log expression ratio of relapse vs non-relapse cases to assess enrichment of CMS/CRIS subtypes. X-axis shows normalised enrichment score (NES), and the adjusted $p$ value is indicated on the colour bar, from red (significant) to blue (not significant). padj, adjusted $p$ value (computed and corrected for multiple testing using the Benjamini–Hochberg procedure). **f** Hallmark gene set enrichment analysis of TGFβ signalling in relapse cases as compared to non-relapse using fast gene set enrichment analysis (fgsea) method. TGFβ_Up gene set from[22]. **g** Correlation analysis between transcriptome-based fibroblast scores (x-axis) and sample-level gsea hallmark TGFβ signalling scores (y-axis) in relapse (red) and non-relapse (blue) pT1 cohort. Two-sided $P = 0.6$, rho = Spearman correlation coefficient. Grey region shows the 95% CI for predictions from (a). **h** Enrichment of cancer-associated fibroblast (CAF), left, and stromal (right) signature gene sets in relapse cases compared to non-relapse samples using fgsea method; Benjamini–Hochberg FDR <0.2 was considered as significant. NES normalised enrichment score, FDR false discovery rate, padj adjusted $p$ value (computed and corrected for multiple testing using the Benjamini–Hochberg procedure).

data demonstrate that TGFβ/ALK5 activation in the intestinal epithelium is SMAD4-dependent.

As TGFβ/ALK5 cytostatic signalling is thought to be critical for resolving the proliferative phase during the regeneration of the colonic mucosa following injury[20,39], we sought to ascertain whether

activation of this pathway would promote intestinal regeneration. To ablate the intestinal epithelium and induce subsequent regeneration, we exposed *VilCre<sup>ER</sup>;Alk5<sup>CA</sup>* mice to a single dose of γ-irradiation (10 Gy), 4 days after tamoxifen injection. Compared with tamoxifen-treated *VilCre<sup>ER</sup>;Alk5<sup>+/+</sup>* controls, the intestinal crypts of *VilCre<sup>ER</sup>;Alk5<sup>CA</sup>*

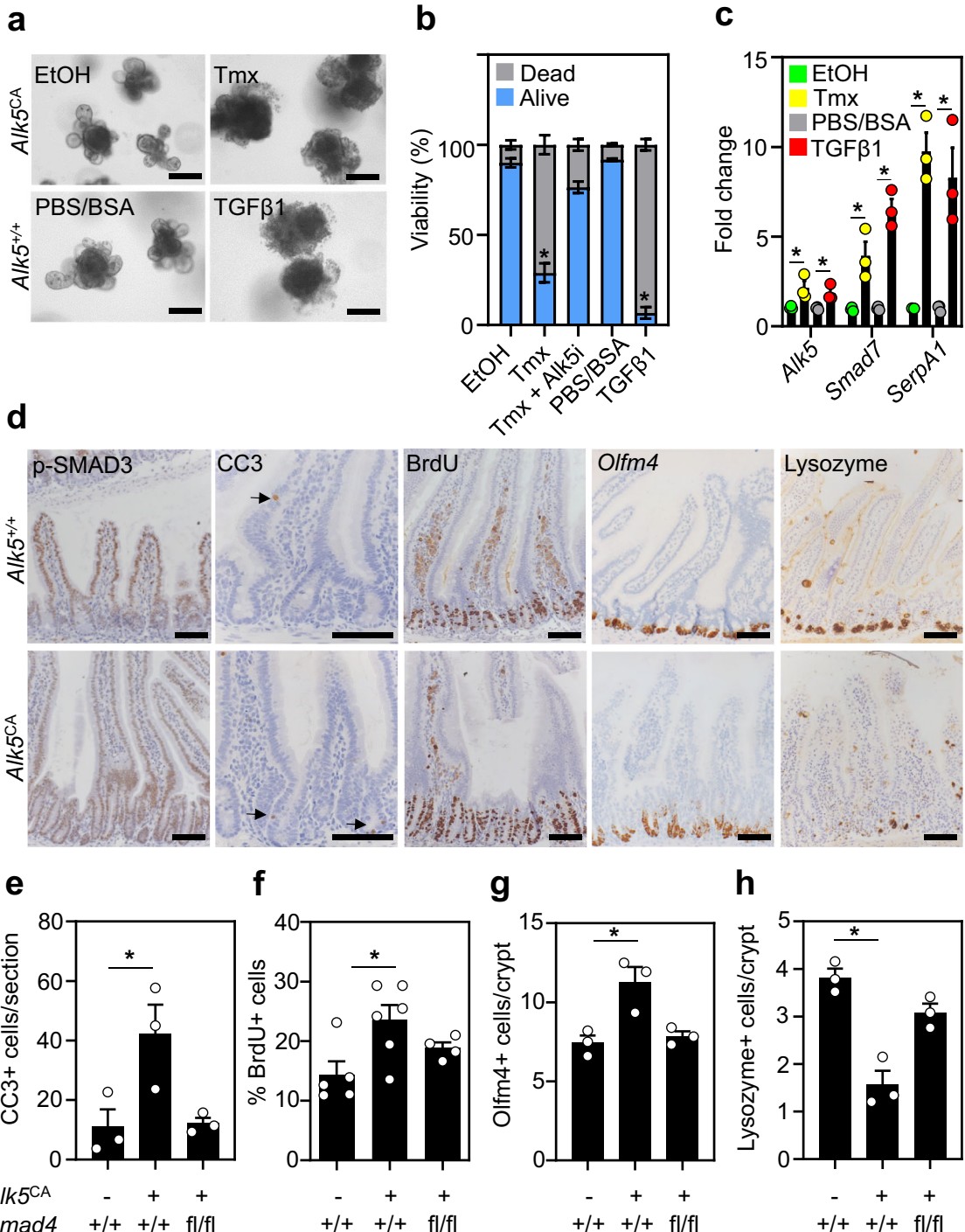

**Fig. 2 | Epithelial cell-intrinsic activation of TGFβ signalling is tolerated in vivo, but not in vitro. a** Representative images of small intestinal organoids derived from uninduced *VilCre^ER;Alk5^CA* (*Alk5^CA*) and *VilCre^ER;Alk5^+/+* (*Alk5^+/+*) mice, following 5 days of in vitro treatment with vehicle control (EtOH for 4-hydroxytamoxifen; PBS/BSA for TGFβ1), 4-hydroxytamoxifen (Tmx, top right panel) or TGFβ1 (bottom right panel). *n* = 3 organoid lines per group. Treatments were repeated twice. Scale bar, 50 μm. **b** Relative viability (dead; grey vs alive; blue) of *Alk5^CA* and *Alk5^+/+* organoids measured 5 days after the indicated treatments. *Alk5^CA* organoids were treated with EtOH, Tmx, or Tmx plus ALK5i, whereas *Alk5^+/+* organoids were treated with PBS/BSA or TGFβ1. Representative images of treated organoids, with or without ALK5i, are shown in Supplementary Fig. 2a. *n* = 3 biological replicates per group, each in technical triplicate. Data were ±s.e.m; *P* = 0.05, one-tail

Mann–Whitney *U*-test. **c** qPCR for TGFβ-target genes expressed by organoids described in **a**. *n* = 3 biological replicates per group; EtOH (green), tamoxifen (yellow), PBS/BSA (grey), TGFβ1 (red) each in technical triplicate. Data were ±s.e.m; *P* = 0.05, one-tail Mann–Whitney *U*-test. **d** Representative IHC (p-SMAD3, cleaved caspase-3 (CC3), BrdU and lysozyme) and ISH (*Olfm4*) in the small intestine of *VilCre^ER;Alk5^+/+* (*Alk5^+/+*) and *VilCre^ER;Alk5^CA* (*Alk5^CA*) mice 4 days post-tamoxifen induction. Arrows indicate CC3-positive apoptotic cells. Scale bar, 100 μm. **e–h** Quantification of small intestinal epithelial cells stained for cleaved caspase-3 (CC3) (**e**), BrdU (**f**), *Olfm4* (**g**), and lysozyme (**h**) cells from mice of the indicated genotypes 4 days post-tamoxifen induction. *n* = 3 mice per group and *P* = 0.05, except for BrdU scoring; *n* = 5 *Alk5^+/+;Smad4^+/+*, *n* = 6 *Alk5^CA;Smad4^+/+* and *n* = 4 *Alk5^CA;Smad4^fl/fl* mice, *P* = 0.03. All data were ±s.e.m; two-tail Mann–Whitney *U*-test.

mice displayed enhanced regenerative capacity 72 h post irradiation, accompanied by high levels of BrdU incorporation and p-SMAD3 staining (Supplementary Fig. 2d, e). These data suggest that the activation of epithelial TGFβ/ALK5 signalling during regeneration drives cell proliferation, which accelerates recovery following injury.

### Acute activation of TGFβ signalling drives apoptosis of Apc-mutant cells in vitro and in vivo

*Apc*-mutant cells are hypersensitive to TGFβ-induced apoptosis in vitro[40], and we have recently shown that the secreted Wnt-agonist R-spondin slows intestinal tumourigenesis in *Apc*[Min/+] mice by activating TGFβ signalling[41]. Given *APC* is mutated in up to 80% of CRC[27], we sought to determine whether constitutive activation of epithelial TGFβ/ALK5 receptor signalling would promote or suppress tumour development induced by *Apc* loss and whether such pathway crosstalk could be recapitulated in organoids. To address this, we crossed *Vil-Cre*[ER]*;Alk5*[CA] mice with *Apc*[fl/fl] mice[42], generating *VilCre*[ER]*;Apc*[fl/fl]*;Alk5*[CA] offspring. We first confirmed that constitutive activation of TGFβ/ALK5 signalling impaired the growth of *VilCre*[ER]*;Apc*[fl/fl]*;Alk5*[CA] cells, leading to pronounced organoid atrophy and diminished viability within 5 days of in vitro induction, compared with vehicle-treated cultures (Supplementary Fig. 3a, b). To test whether sensitivity to TGFβ signalling is specific to *Apc*-mutant cells or a general property of cells with aberrant Wnt activation, we targeted mutant *CTNNB1* (gene encoding β-catenin), specifically exon 3 which encodes serine and theronine sites phosphorylated by GSK3-β to promote β-catenin degradation, to the gut epithelium to drive constitutive Wnt/β-catenin pathway activation[43]. Consistent with the phenotype observed in *Apc*-mutant organoid cultures subjected to constitutive TGFβ/ALK5 signalling, intestinal organoids derived from *VilCre*[ER]*;β-cat*[Ex3/Ex3] mice 4 days post-tamoxifen induction underwent rapid cell death and reduced viability following treatment with TGFβ1 (Supplementary Fig. 3c). Together, these data demonstrate cells with hyperactivated Wnt signalling are exquisitely sensitive to TGFβ/ALK5 signalling.

Next, we injected mice with tamoxifen to induce constitutive activation of TGFβ/ALK5 signalling in vivo and examined intestinal tissues 3 days post-tamoxifen induction. At this timepoint, intestinal crypts from *VilCre*[ER]*;Apc*[fl/fl]*;Alk5*[CA] mice had significantly fewer proliferating epithelial cells and a robust induction of cell death compared with the hyper-proliferative crypts from V*ilCre*[ER]*;Apc*[fl/fl]*;Alk5*[+/+] mice (Fig. 3a and Supplementary Fig. 3d). Interestingly, at a later timepoint (4 days post-tamoxifen), *VilCre*[ER]*;Apc*[fl/fl]*;Alk5*[CA] intestinal crypts displayed increased proliferation and reduced cell death, compared with V*ilCre*[ER]*;Apc*[fl/fl]*;Alk5*[+/+] counterparts, raising the possibility of selection against the constitutive activation of ALK5 signalling (and the *Alk5*[CA] transgene) in *Apc*-mutant cells (Fig. 3a, b). To test this further, we used a custom-design RNAscope probe to detect expression of the *Alk5*[CA] transgene and stained intestinal tissue from *VilCre*[ER]*;Apc*[fl/fl]*;Alk5*[CA] mice 3 (d3) and 4 (d4) days post-tamoxifen injection. Consistent with our hypothesis, we observed a striking reduction in *Alk5*[CA] transgene expression at d4 compared with d3 (Supplementary Fig. 3e). Moreover, organoids, established from *VilCre*[ER]*;Apc*[fl/fl]*;Alk5*[CA] mice 4 days after in vivo induction, grew comparably to V*ilCre*[ER]*;Apc*[fl/fl]*;Alk5*[+/+] cells and exhibited significantly reduced expression of the *Alk5*[CA] transgene and TGFβ downstream targets, compared with organoids induced in vitro (Supplementary Fig. 3f, g). This indicates that there is a strong negative selection against *Alk5*[CA]-positive cells and/or downregulation of the *Alk5*[CA] transgene.

To overcome the negative selection against prolonged *Alk5*[CA] transgene expression, we used the ALK5-inhibitor (ALK5i) reasoning that temporal inhibition of *Alk5*[CA] receptor signalling, followed by acute resumption of *Alk5*[CA] activity, may recapitulate the cytotoxicity observed in vitro (Supplementary Fig. 3b). For this, we transiently treated tamoxifen-induced *VilCre*[ER]*;Apc*[fl/fl]*;Alk5*[CA] mice with ALK5i (50 mg/kg) for 2 days and harvested small intestinal tissues one day

post release from ALK5i treatment (Fig. 3c). Compared with vehicle-treated controls, ALK5i-treated *VilCre*[ER]*;Apc*[fl/fl]*;Alk5*[CA] mice displayed robust and more extensive activation of TGFβ/ALK5 signalling, following cessation of treatment, as shown by p-SMAD3 staining and *Alk5*[CA] transgene expression (Fig. 3d), which was accompanied by a significant reduction in cell proliferation and the induction of apoptosis (Fig. 3e).

Consistent with these data, we found that tumours arising in aging tamoxifen-induced *VilCre*[ER]*;Apc*[fl/+]*;Alk5*[CA] mice had low levels of *Alk5*[CA] transcripts, whilst adjacent normal tissue maintained expression of the *Alk5*[CA] transgene (Supplementary Fig. 3h). Additionally, we observed no significant differences in survival and tumourigenesis, and the expression of TGFβ targets remained low in the tumour tissue suggesting that all the tumours that arose had escaped recombination (Supplementary Fig. 3i–k).

Together, these data indicate that epithelial cell-intrinsic TGFβ/ALK5 signalling is poorly tolerated by *Apc*-mutant cells, acting as a tumour suppressor, in contrast to our findings in epithelial regeneration, where its functions are pro-reparative and even pro-proliferative. This observation is consistent with the commonplace coincident mutation of *APC* and *SMAD4* in CRC, but suggests that early disseminating tumours require additional hits to enable the epithelium to tolerate TGFβ-pathway activation.

### Wnt, MAPK and TGFβ signalling co-operate to promote intestinal tumourigenesis

Sequencing of the early disseminating lesions in *APC*-mutant patients indicated that aggressive traits were associated with an enrichment for oncogenic *KRAS* mutations and transcriptomic activation of KRAS-associated gene programmes (Fig. 1c, d). For this reason, we interbred *VilCre*[ER]*;Apc*[fl/+]*;Alk5*[CA] mice with mice bearing oncogenic KRAS[G12D] (*Kras*[G12D/+]) mutation[44], generating *VilCre*[ER]*;Apc*[fl/+]*;Kras*[G12D/+]*;Alk5*[CA] offspring. Compared with *VilCre*[ER]*;Apc*[fl/+]*;Kras*[G12D/+]*;Alk5*[+/+] animals, *VilCre*[ER]*;Apc*[fl/+]*;Kras*[G12D/+]*;Alk5*[CA] mice succumbed significantly more rapidly to their tumour burden (Fig. 4a–c), although we found no appreciable differences in the immune cell infiltrates or stroma (Supplementary Fig. 4a). Interestingly, compared with controls, *VilCre*[ER]*;Apc*[fl/+]*;Kras*[G12D/+]*;Alk5*[CA] mice harboured a reduced large-intestinal tumour burden (Supplementary Fig. 4b, c), suggesting that the rapid development of small intestinal tumours elicited morbidity before colonic tumours could fully establish. Importantly, *VilCre*[ER]*;Apc*[fl/+]*;Kras*[G12D/+]*;Alk5*[CA] tumours exhibited high expression of the *Alk5*[CA] transgene and prominent p-SMAD3 staining (Fig. 4d, e and Supplementary Fig. 4a), in contrast to APC-deficient tumours lacking *Kras* mutation (Supplementary Fig. 3h, j). As expected, the tumour epithelium in these mice also exhibited activation of the MAPK- and Wnt-signalling pathways (Fig. 4e). Together, these findings are consistent with the switch of epithelial TGFβ signalling from a tumour suppressor to a tumour promoter, in the context of concurrent *Apc* and *Kras* mutations, and underscore the fact that the output of TGFβ signalling is highly contextual.

### Oncogenic KRAS protects cells with active TGFβ/ALK5 signalling from apoptosis

To test whether oncogenic KRAS is sufficient to render *Apc*-mutant cells able to tolerate the activation of TGFβ signalling, we dissociated *VilCre*[ER]*;Apc*[fl/fl]*;Kras*[G12D/+] organoids into single-cells and treated for 5 days with doses of TGFβ that kill *Apc*-mutant organoids. We found that prior to passage, the majority of *VilCre*[ER]*;Apc*[fl/fl]*;Kras*[G12D/+] organoids were refractory to TGFβ treatment (Supplementary Fig. 5a, b−P0 data) and exhibited a marked induction of TGFβ downstream targets (*Smad7*, *SerpA1*) but not of apoptotic genes (Supplementary Fig. 5c). However, following mechanical passage (into single-cells), the viability of TGFβ-treated *VilCre*[ER]*;Apc*[fl/fl]*;Kras*[G12D/+] cells was significantly compromised compared with vehicle-treated cells (Supplementary Fig. 5b −P1 data), which is comparable to the sensitivity of TGFβ-treated

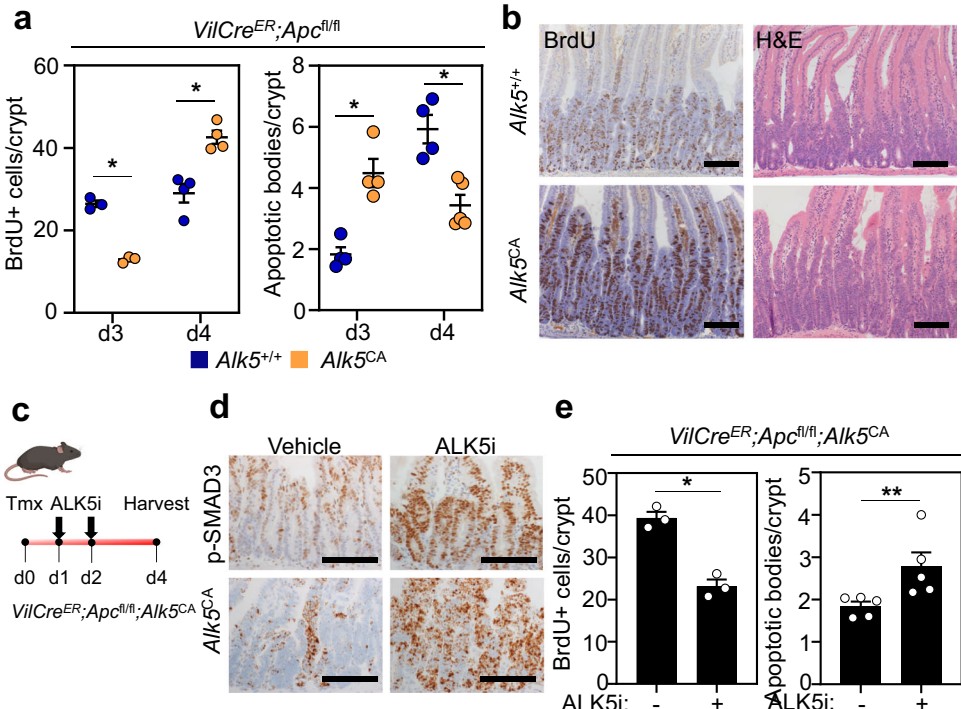

**Fig. 3 | Apc-mutant cells are sensitive to cell-intrinsic TGFβ signalling.**
**a** Quantification of BrdU-positive (left) and apoptotic cells (right) in *VilCre^ER;Apc^fl/fl;Alk5^+/+* (blue) and *VilCre^ER;Apc^fl/fl;Alk5^CA* (orange) small intestinal crypts 3 and 4 days post-tamoxifen induction. *n* = 4 mice per group, except *n* = 3 mice per group for d3 BrdU data. Data were ±s.e.m; *P = 0.05, one-tail Mann–Whitney *U*-test. **b** Representative BrdU and H&E staining of small intestinal tissue from *VilCre^ER;Apc^fl/fl;Alk5^+/+* and *Vil-Cre^ER;Apc^fl/fl;Alk5^CA* mice 4 days post-tamoxifen induction. Scale bar, 100 μm. **c** Schematic illustrating the treatment timeline and tissue analysis from *VilCre^ER;Apc^fl/fl;Alk5^CA* mice.

Tmx tamoxifen, ALK5i ALK5-inhibitor. **d** Representative IHC (p-SMAD3) and ISH (*Alk5^CA*) staining on *VilCre^ER;Apc^fl/fl;Alk5^CA* mice following vehicle or ALK5i treatment. Scale bar, 200 μm. **e** Quantification of BrdU-positive (left) and apoptotic cells (right) from *Vil-Cre^ER;Apc^fl/fl;Alk5^CA* mice following vehicle or ALK5i treatment. *n* = 3 mice per group, except for scoring of apoptotic bodies where *n* = 5 mice per group. Data were ±s.e.m. *P = 0.05, **P = 0.008; one-tail Mann–Whitney *U*-test for BrdU; two-tail Mann–Whitney *U*-test for apoptosis.

human *APC.KRAS*-mutant colon organoids[45]. We next generated *Vil-Cre^ER;Apc^fl/fl;Kras^G12D/+;Alk5^CA* organoids and induced Cre-mediated recombination in vitro, which was again accompanied by a marked induction of TGFβ signalling but not of apoptosis (Supplementary Fig. 5d, e). Importantly, there was no decrease in organoid viability following ALK5^CA induction in vitro (Supplementary Fig. 5f), consistent with the notion that oncogenic KRAS enables *Apc*-mutant cells to tolerate TGFβ signalling in vitro. We next induced *VilCre^ER;Apc^fl/fl;Kras^G12D/+;Alk5^CA* mice in vivo and harvested intestinal tissue 3 days post induction. The corresponding tissues did not show an induction of apoptosis and had markedly hyperplastic crypts. To assess if acute TGFβ activation could drive apoptosis, we treated tamoxifen-induced *VilCre^ER;Apc^fl/fl;Kras^G12D/+;Alk5^CA* mice for 2 days with ALK5i and subsequently released them from inhibitor treatment (Fig. 5a). Unlike *VilCre^ER;Apc^fl/fl;Alk5^CA* mice (Fig. 3f–h), we did not observe changes in cell proliferation and apoptosis in ALK5i-treated *VilCre^ER;Apc^fl/fl;Alk5^CA* mice (Fig. 5b), despite the increased number of *Alk5*-positive cells (Fig. 5a). We next assayed both TGFβ and MAPK signalling and observed robust expression of TGFβ-responsive transcript *Itgβ6* and phosphorylated ERK (p-ERK), respectively, in tamoxifen-induced *VilCre^ER;Apc^fl/fl;Kras^G12D/+;Alk5^CA* mice compared with *VilCre^ER;Apc^fl/fl;Kras^G12D/+;Alk5^+/+* mice (Fig. 5c, d).

We have previously shown that *VilCre^ER;Apc^fl/fl;Kras^G12D/+* mice are resistant to MAPK inhibition[13], consistent with the failure of MEK inhibitors in clinical trial of patients with *KRAS*-mutant CRC[46]. Therefore, to test whether constitutive ALK5-dependent TGFβ signalling confers sensitivity to MEK inhibition in this genetic context, we treated *VilCre^ER;Apc^fl/fl;Kras^G12D/+;Alk5^CA* mice with a selective MEK1/2 inhibitor (selumetinib/AZD6244 (MEKi); Supplementary Fig. 5g). Compared with vehicle-treated controls, MEKi-treated mice displayed

significantly reduced cell proliferation and an induction of apoptosis (Fig. 5e, f), which were paralleled by increased *Alk5^CA* expression (Supplementary Fig. 5h). To overcome any confounding effects of MEK1/2-inhibited stroma, we induced recombination in *VilCre^ER; Apc^fl/fl;Kras^G12D/+;Alk5^CA* organoids in vitro and supplemented the culture medium with MEKi. Similar to our in vivo findings, treatment with MEKi reduced organoid cell viability and increased cell death (Supplementary Fig. 5d–f). Collectively, these data confirm that the activation of MAPK signalling renders *Kras*-mutant cells able to withstand pro-apoptotic cell-intrinsic TGFβ/ALK5 signalling, permitting their retention and survival in the intestine.

## Epithelial TGFβ/ALK5 signalling confers transcriptional features of aggressive CRC

The mouse model that we have developed in this study demonstrates that epithelial TGFβ signalling is tolerated and promotes intestinal tumourigenesis in concert with mutations in *Apc* and *Kras*. We next examined whether our model recapitulates the TGFβ-high CMS4 and CRIS-B subtypes of human CRC and, more importantly, whether it resembles the *born to be bad*, early disseminating lesions characterised by high TGFβ signalling.

For this, we generated transcriptome-wide expression profiles from mouse small intestinal tissue, with the aim of developing a transcriptional classifier for epithelial-specific TGFβ/ALK5 activation. We performed differential analysis of gene expression data from *Vil-Cre^ER;Alk5^CA* and *VilCre^ER;Alk5^+/+* mice 4 days post-tamoxifen induction, designated ALK5^CA-like and WT-like (non-TGFβ/Alk5-activated) respectively. After filtering to enrich for genes with potential predictive value, we identified 60 genes that could robustly discriminate between WT-like and ALK5^CA-like mouse tissue (Fig. 6a). We next

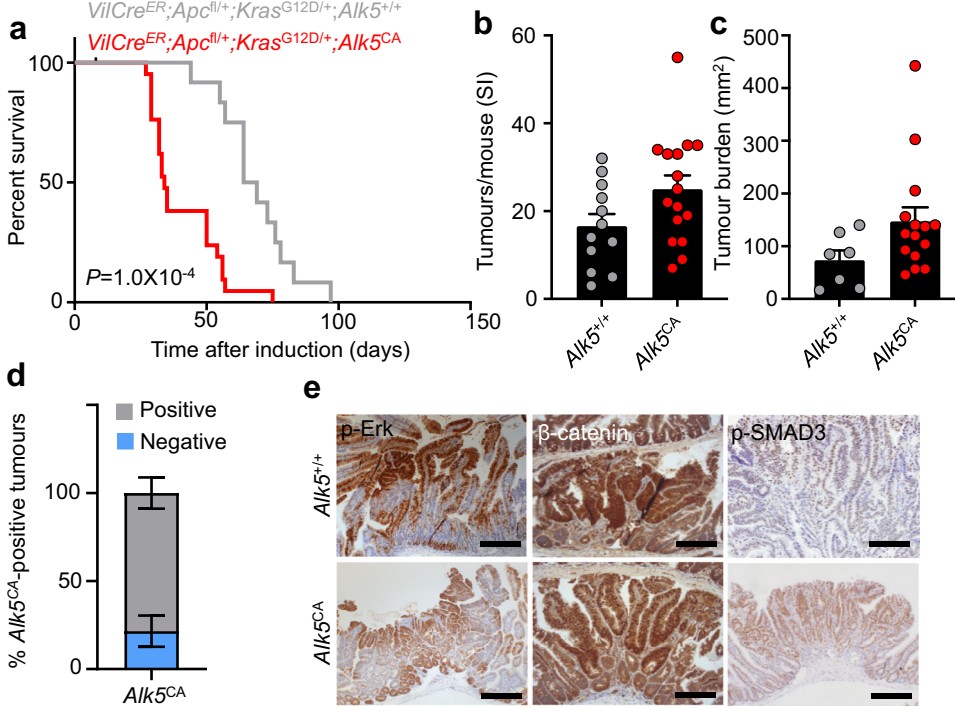

**Fig. 4 | Epithelial TGFβ/ALK5 signalling synergises with Wnt and MAPK signalling to accelerate tumourigenesis. a** Survival plot for *VilCre*[ER]*;Apc*[fl/+]*;Kras*[G12D/+]*;Alk5*[+/+] (n = 12, grey) and *VilCre*[ER]*;Apc*[fl/+]*;Kras*[G12D/+]*;Alk5*[CA] (n = 22, red) mice aged until clinical endpoint following tamoxifen induction. P = 1.0 × 10⁻⁴, log-rank test. **b**, **c** Small intestinal (SI) tumour number (**b**) and burden (**c**) per mouse from mice described in **a**. n = 12 *Alk5*[+/+] mice (grey), n = 16 *Alk5*[CA] mice (red) for tumour number dataset; n = 7 *Alk5*[+/+] mice (grey), n = 15 *Alk5*[CA] mice (red) for tumour burden dataset. Data were ±s.e.m; P = 0.064 (**b**), P = 0.063 (**c**), two-tail Mann−Whitney *U*-test. **d** Relative percentage of small intestinal tumours from *VilCre*[ER]*;Apc*[fl/+]*;Kras*[G12D/+]*;Alk5*[CA] (*Alk5*[CA]) mice positive (grey) or negative (blue) for *Alk5*[CA] expression (ISH). n = 4 mice. Data were ±s.e.m. **e** Representative IHC for p-ERK, β-catenin and p-SMAD3 on tumour tissue from mice described in **a**. Scale bar, 200 μm.

applied our 60-gene classifier to our pT1 cohort enriched for aggressive, early disseminating tumours. Importantly, we observed that 90% of relapsing tumours were associated with epithelial-specific TGFβ activation, whereas only 29% of non-relapsing tumours aligned to ALK5[CA]-like transcriptional features (Fig. 6b). Given that prognostic factors in later-stage disease failed to discriminate between relapse and non-relapse cases in our pT1 cohort (Fig. 1), we next sought to test the wider applicability of our pT1 risk model in stratifying stage II CRC cases based on risk-of-relapse. Using a multiplexed immuno-fluorescence method with single cell-level quantification[47], we assessed the expression of key components in the mTOR and TGFβ signalling pathways by single-cell staining specifically of epithelial tumour cells within samples from a cohort of 282 stage II CRCs. This analysis revealed 4 distinct clusters (Supplementary Fig. 6a), with the expression patterns in cluster 4 aligning with relapse in our original T1 cohort; namely, elevated TGFβ, p-mTOR and p-S6, alongside repressed 4EBP1 (Fig. 6c and Supplementary Fig. 6b, c). Ranking of patients according to the proportion of single cells associating with cluster 4, followed by relapse-free survival (RFS) analysis, revealed that the epithelial signalling pathways associated with pT1 relapse could also identify a subgroup of stage II CRCs with a significantly worse prognosis (HR = 3.37; 95% CI, 1.5-7.55; Fig. 6d and Supplementary Table 2).

Consistent with the signalling pathways upregulated in aggressive T1 tumours (Fig. 1), *VilCre*[ER]*;Apc*[fl/fl]*;Kras*[G12D/+]*;Alk5*[CA] mice exhibit a significant enrichment of TGFβ signalling (Fig. 6e; Fessler et al. TGFβ-responsive gene signature and Hallmark TGFβ-pathway classifier[22,28]), alongside upregulation of KRAS and mTORC1 signalling (Fig. 6f; Hallmark KRAS and mTORC1 pathway gene sets[28]). Additionally, consistent with the fact that our model harbours epithelial-intrinsic−rather than stromal−activation of TGFβ signalling, GSEA analysis showed an enrichment of signalling signatures associated with CRIS-B (Fig. 6g), and

to a lesser extent CRIS-A, compared with the remaining intrinsic subtypes (Fig. 6h and Supplementary Fig. 6d). Further single-sample GSEA (ssGSEA) comparisons of the CRIS-B signature with the gene expression profiles of our *VilCre*[ER]*;Apc*[fl/fl]*;Kras*[G12D/+]-driven models, harbouring no additional mutation (AK), constitutive ALK5-dependent TGFβ activation (AKA[CA]), or *Alk5* gene knockout (AKAKO), demonstrate that the acquisition of a CRIS-B gene expression signature is mostly associated with epithelial cell-intrinsic TGFβ/ALK5-pathway activity, with little-to-no contribution from deregulated Wnt or MAPK signalling (Fig. 6i).

To better understand how Wnt and MAPK signalling might synergise with TGFβ/ALK5 activation to drive these pro-oncogenic transcriptional programmes and accelerate intestinal tumourigenesis, we interrogated RNA-sequencing data to identify preferentially deregulated cellular processes and gene expression programmes in *VilCre*[ER]*;Apc*[fl/fl]*;Kras*[G12D/+]*;Alk5*[CA] tissue (Supplementary Fig. 6e, f). Of the gene programmes differentially expressed, we observed a significant enrichment of multiple genes involved in TGFβ signal transduction (*Tgfbr1/Alk5*, *Fos*, *Smurf1*, *Smad7*) and growth-factor signalling (*Ret*, *Vegfa*, *Egfr*, *Cblb*, *Pdgfb*) (Fig. 6j), which are often deregulated in CRC[48–50]. To assess whether these gene programmes are dependent on canonical TGFβ signalling, we interbred *Smad4*[fl/fl] mice with *VilCre*[ER]*;Apc*[fl/fl]*;Kras*[G12D/+]*;Alk5*[CA] (AKA[CA]) mice to generate *VilCre*[ER]*;Apc*[fl/fl]*;Kras*[G12D/+]*;Alk5*[CA]*;Smad4*[fl/fl] (AKA[CA]S) mice and examined the consequences of *Smad4* loss on the phenotypic and transcriptional output of TGFβ/ALK5 signalling. In parallel, we crossed *Alk5*[fl/fl] mice with *VilCre*[ER]*;Apc*[fl/fl]*;Kras*[G12D/+] mice to generate *VilCre*[ER]*;Apc*[fl/fl]*;Kras*[G12D/+]*;Alk5*[fl/fl] (AKAKO) mice, enabling conditional ablation of ALK5-dependent TGFβ signalling in the intestinal epithelium. Using these two models, we found that the expression of genes involved in growth-factor signalling, enriched following TGFβ/ALK5 activation, is dependent on both ALK5 and SMAD4, suggesting that they are canonical TGFβ-target genes (Fig. 6j).

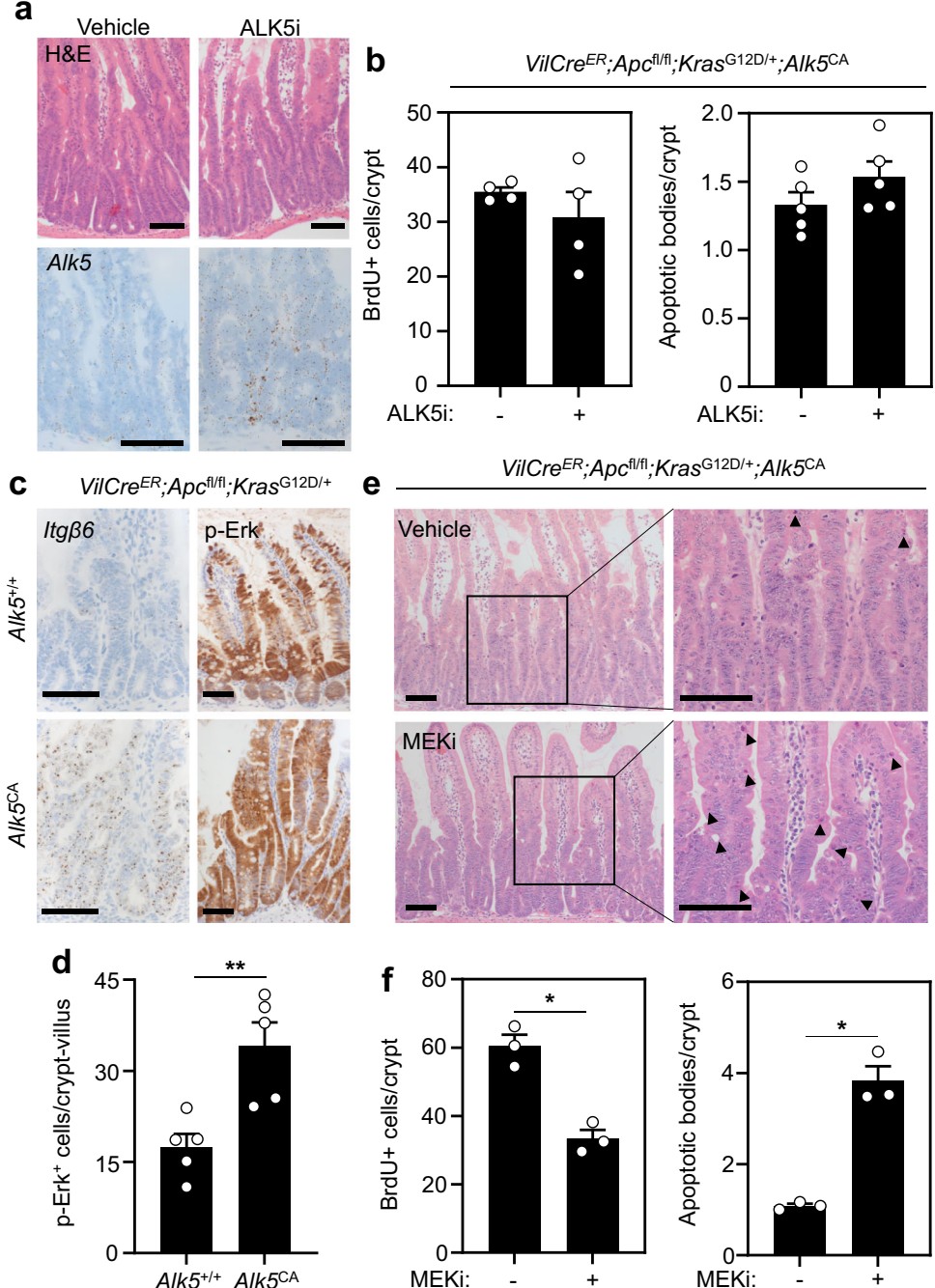

**Fig. 5 | Oncogenic KRAS-driven MAPK signalling confers tolerance to epithelial TGFβ activation. a** Representative H&E and *Alk5* ISH of small intestinal tissue from *VilCre^ER;Apc*^fl/fl*;Kras*^G12D/+*;Alk5*^CA mice 3 days post-tamoxifen and daily treatment with vehicle or ALK5-inhibitor (ALK5i). Scale bar, 100 µm. **b** Quantification of BrdU-positive (left) and apoptotic (right) cells in mice described in **a**. *n* = 4 mice per group (BrdU scoring); *n* = 5 mice per group (apoptosis scoring). Data were ±s.e.m; *P* = 0.68 (BrdU), *P* = 0.15 (apoptosis); two-tail Mann–Whitney *U*-test. **c** Representative *Itgβ6* ISH and p-ERK IHC on mice of the indicated genotypes 3 days post-tamoxifen. Scale bars, 100 µm. **d** Quantification of p-ERK-positive cells per crypt-villus unit in mice

described in **c**. *n* = 5 per group. Data were ±s.e.m; *P* = 0.003. Two-tail Mann–Whitney *U*-test. **e** H&E staining of small intestinal tissue from *VilCre^ER; Apc*^fl/fl*;Kras*^G12D/+*;Alk5*^CA mice 3 days post-tamoxifen and daily treatment with vehicle or MEK1/2 inhibitor (MEKi). Right panels show magnification of boxed areas in left panels, with apoptotic cells indicated by black arrowheads. Scale bar, 100 µm. **f** Quantification of BrdU-positive (left) and apoptotic (right) cells in mice described in **e**. *n* = 3 mice per group. Data were ± s.e.m. **P* = 0.05; one-tail Mann–Whitney *U*-test.

## Aberrant TGFβ signalling sensitises epithelial tumour cells to growth-factor receptor inhibition

Our data suggest that there are two independent, yet cooperative, pathways driving MAPK activation, one via oncogenic KRAS and the other via epithelial-intrinsic TGFβ signalling. Following TGFβ activation, the engagement of multiple growth-factor receptor pathways,

eliciting EGFR, RET and PDGFβ signal transduction, would be expected to mimic tissue regeneration and further augment MAPK-pathway activation. Moreover, a dependence on growth-factor receptor signalling may provide a tractable therapeutic vulnerability in CRCs with high epithelial cell-intrinsic TGFβ signalling that can be targeted with clinically relevant inhibitors.

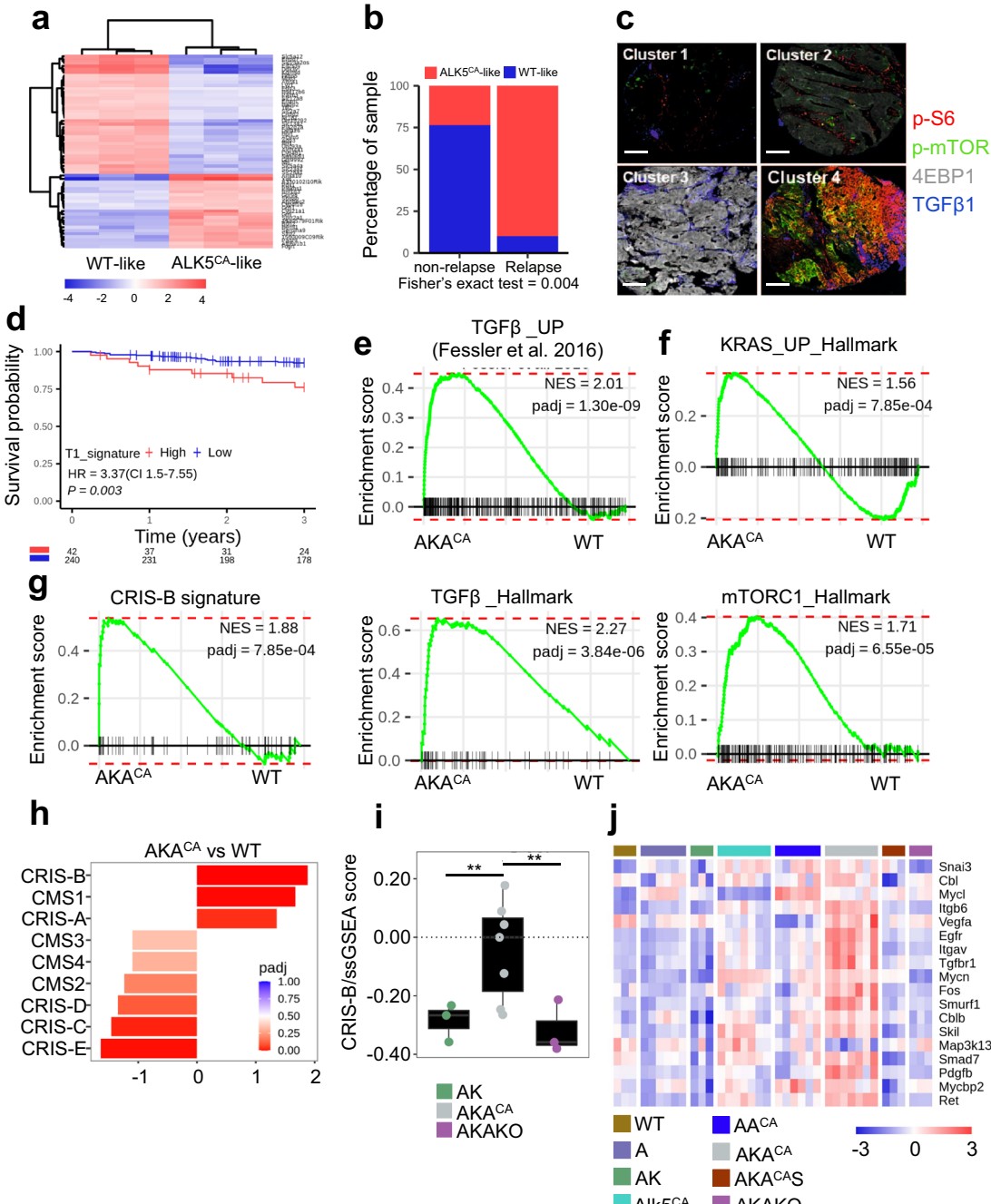

**Fig. 6 | Mice with intestinal epithelial TGFβ signalling mimic transcriptional features of human born to be bad pT1 tumours. a** Heatmap of 60 genes with predictive value obtained from prediction analysis for microarrays (PAMR) using data from *VilCre^ER;Alk5^CA* (ALK5^CA-like) and non-TGFβ/Alk5-activated (WT-like) mice 4 days post-tamoxifen. *n* = 3 mice per genotype. **b** Proportion of relapse and non-relapse pT1 patients enriched with ALK5^CA-like (red) or WT-like (blue) signature. Two-sided Fisher's exact test using fisher.test function in stats R package. **c** Pseudocolour overlay images of representative patient tumours from each cluster stained for p-S6, p-mTOR, 4EBP1 and TGFβ1. Marker expression was assessed using multiplexed immunofluorescence and single-cell staining of epithelial tumour cells within a cohort of stage II CRC (*n* = 282). Scale bar, 200 μm. **d** Proteins (p-S6, 4EBP1, TGFβ1 and p-mTOR) associated with relapse in the pT1 cohort were used to stratify stage II CRC patients into two groups, high (red) and low (blue) expression (*P* = 0.003; log-rank test). **e, f** GSEA of TGFβ (**e**) and KRAS and mTORC1 (**f**) signatures on shrunken log expression ratio of *VilCre^ER;Apc^fl/fl;Kras^G12D/+;Alk5^CA* (AKA^CA; *n* = 7) vs wild type (WT; *n* = 3) intestinal tissue 3 days post-tamoxifen using fgsea method. **g** CRIS-B subtype enrichment analysis on mice described in **e**. **h** Group-wise GSEA performed using fgsea R package on the log expression ratio of AKA^CA vs

WT mouse intestinal tissue to assess enrichment of CMS/CRIS subtypes. X-axis shows normalised enrichment score (NES), and the adjusted *p* value is indicated on the colour bar, from red (significant) to blue (not significant). padj adjusted *p* value (computed and corrected for multiple testing using the Benjamini–Hochberg procedure). **i** CRIS-B signature single-sample GSEA analysis on *VilCre^ER; Apc^fl/fl;Kras^G12D/+* (AK, green), *VilCre^ER;Apc^fl/fl;Kras^G12D/+;Alk5^CA* (AKA^CA, grey) and *Vil-Cre^ER;Apc^fl/fl;Kras^G12D/+;Alk5^fl/fl* (AKAKO, purple) tissue 3 days post-tamoxifen. AK, AKAKO *n* = 3 mice, AKA^CA *n* = 7 mice. Statistical significance determined by two-sided *t*-test using t.test function in stats R package (4.2.0). Horizontal line represents median values, boxes indicate the inter-quartile range and bars denote the maximum and minimum values. ***$p \le 0.001$; **$p \le 0.01$; *$p \le 0.05$. **j** Heatmap of genes associated with growth-factor and TGFβ signalling differentially expressed in intestinal tissue harvested 3 days post-tamoxifen injection from mice of the indicated genotypes. Wild-type (WT, *n* = 3, mustard), *VilCre^ER;Apc^fl/fl* (A, *n* = 6, lavender), *VilCre^ER;Apc^fl/fl;Kras^G12D/+* (AK, *n* = 3, green), *VilCre^ER;Alk5^CA* (Alk5^CA, *n* = 7, aqua), *Vil-Cre^ER;Apc^fl/fl;Alk5^CA* (AA^CA, *n* = 6, blue), *VilCre^ER;Apc^fl/fl;Kras^G12D/+;Alk5^CA* (AKA^CA, *n* = 7, grey), *VilCre^ER;Apc^fl/fl;Kras^G12D/+;Alk5^CA;Smad4^fl/fl* (AKA^CAS, *n* = 3, brown) and *VilCre^ER; Apc^fl/fl;Kras^G12D/+;Alk5^fl/fl* (AKAKO, *n* = 3, purple).

Given the activation of multiple growth-factor receptor pathways in tumours with elevated epithelial cell-intrinsic TGFβ signalling, we first treated tamoxifen-induced $VilCre^{ER}$;$Apc^{fl/+}$;$Kras^{G12D/+}$;$Alk5^{CA}$ mice with vandetanib/ZD6474, a small-molecule tyrosine kinase inhibitor selective for VEGFR, RET, and EGFR, until clinical symptoms manifested. Compared with vehicle-treated controls, ZD6474-treated mice exhibited a significant extension in survival (Fig. 7a), which was accompanied by reduced angiogenesis (assessed by CD31 immunostaining), and diminished MAPK (p-ERK) and RET signalling (Fig. 7b and Supplementary Fig. 7a). Moreover, ZD6474-treated mice had significantly fewer small intestinal tumours, demonstrating the functional requirement of VEGFR, RET, and EGFR activity for the optimal growth of intestinal tumours with activated TGFβ/ALK5 signalling (Fig. 7c). Additionally, short-term treatment of $VilCre^{ER}$;$Apc^{fl/fl}$;$Kras^{G12D/+}$;$Alk5^{CA}$ mice with ZD6474 suppressed aberrant cell proliferation and the expansion of $Olfm4^+$ ISCs, and induced apoptosis (Supplementary Fig. 7a–c). Importantly, ZD6474 treatment had no impact on the survival of $VilCre^{ER}$;$Apc^{fl/+}$;$Kras^{G12D/+}$ mice, highlighting the specificity of this drug for tumours harbouring aberrant activation of TGFβ/ALK5 signalling (Supplementary Fig. 7d). Furthermore, the growth and viability of ZD6474-treated $VilCre^{ER}$;$Apc^{fl/fl}$;$Kras^{G12D/+}$;$Alk5^{CA}$ organoids was significantly impaired compared with vehicle-treated controls, underscoring the epithelial cell-intrinsic requirement of this druggable growth-factor module for cells with activated TGFβ/ALK5 signalling (Supplementary Fig. 7e). Crucially, ZD6474 treatment inhibited only the growth of organoids and mouse tumours with simultaneous activation of Wnt, MAPK, and TGFβ/ALK5 signalling (Supplementary Fig. 7f–h), exposing a potential therapeutic vulnerability of tumours harbouring increased growth-factor signalling via TGFβ/ALK5 activation.

Given that ZD6474 is a multi-tyrosine kinase inhibitor that targets VEGFR, RET, and EGFR, and that EGFR signalling plays pivotal roles in colorectal tumour growth, disease progression, and therapy resistance, we sought to determine the extent to which the activation of EGFR might be responsible for accelerating tumourigenesis driven by TGFβ-pathway activation. Indeed, we found that tumour growth could be slowed by treatment with the EGFR-inhibitor sapitinib (AZD8931/EGFRi) specifically in $VilCre^{ER}$;$Apc^{fl/+}$;$Kras^{G12D/+}$;$Alk5^{CA}$ mice but not in $VilCre^{ER}$;$Apc^{fl/+}$;$Kras^{G12D/+}$ mice (Fig. 7d–f and Supplementary Fig. 7d). Interestingly, both ZD6474 and EGFRi monotherapies restored the survival of $VilCre^{ER}$;$Apc^{fl/+}$;$Kras^{G12D/+}$;$Alk5^{CA}$ mice only to a level comparable to that of $VilCre^{ER}$;$Apc^{fl/+}$;$Kras^{G12D/+}$ mice. Given that oncogenic KRAS drives the activation of MAPK signalling, we treated $VilCre^{ER}$;$Apc^{fl/+}$;$Kras^{G12D/+}$;$Alk5^{CA}$ mice with MEKi and found a profound increase in survival compared with targeting EGFR signalling (Fig. 7d–f). Importantly, this increase in the survival of MEKi-treated $VilCre^{ER}$;$Apc^{fl/+}$;$Kras^{G12D/+}$;$Alk5^{CA}$ mice was much greater than we have previously observed with $VilCre^{ER}$;$Apc^{fl/+}$;$Kras^{G12D/+}$ mice[13], demonstrating a higher dependence on MAPK signalling in the presence of epithelial TGFβ activation. Collectively, these data illustrate that the convergence of activated Wnt, MAPK, and TGFβ/ALK5 signalling drives mitogenic and survival pathways that can be targeted therapeutically to slow the progression of intestinal tumours with aggressive behavioural traits.

## Discussion

Consistent with the 'Big Bang' model of colorectal tumour growth[51], recent data have challenged traditional linear tumour progression models, suggesting that early dissemination of metastasis-founder cells can occur at the very earliest stages of CRC development, potentially before current screening methods can routinely detect tumours[7]. However, these studies provide limited evidence of the transcriptional machinery and signal transduction cascades underpinning highly aggressive tumour biology. In the present study, we assemble and molecularly profile tissue from a discovery cohort of pT1 CRCs, enriched for *born to be bad* tumours, providing mechanistic insights into the transcriptional signalling that underpins the highly aggressive phenotype observed in a subset of early-stage, dissemination-prone CRCs. We find that previously validated histology-based prognostic classifiers for later-stage disease are of limited clinical value in our pT1 cohort, but we identify that elevated levels of epithelial cell-intrinsic TGFβ signalling, particularly in the context of concurrent APC and KRAS mutations, are associated with aggressive pT1 biology. Additionally, whereas numerous other signalling pathways are likely to contribute to the aggressive tumour phenotypes observed in patients, transcriptional profiling of our GEMMs identifies epithelial tumour cell-intrinsic TGFβ signalling as a potentially actionable, predictive biomarker in this setting. Accordingly, the defining transcriptomic features of poor-prognosis, dissemination-prone pT1 CRCs can reliably identify the most at-risk patients in a retrospective stage II CRC cohort. With further validation, our signature could inform the management of disease at the earliest, potentially curable stage.

Our study further adds to an ever increasing and complex literature on the dichotomous impact of TGFβ signalling in CRC, with both tumour-suppressive and oncogenic activities ascribed. Indeed, we and others have previously shown that epithelial-specific loss of *Tgfbr1/2* or *Smad4* accelerates tumourigenesis in the context of *Apc* and *Kras* mutations[10,12–14], consistent with key tumour-suppressive roles for epithelial TGFβ signalling in CRC. However, other studies indicate that inactivating TGFβ-pathway mutations in the tumour epithelium drive compensatory TGFβ activation within the TME, fostering tumour development, immune evasion, and disease progression[16,52], whereas other works report that active TGFβ signalling in the tumour epithelium correlates with a pro-invasive EMT-like phenotype and chemotherapy resistance[15,21]. Nevertheless, high levels of TGFβ consistently predict dismal clinical outcome in CRC patients[18,53].

Interestingly, we previously demonstrated that loss of *Tgfbr1/Alk5* promotes the dedifferentiation and tumour-initiating capability of *Apc/Kras*-mutant differentiated intestinal epithelial cells[13]. By contrast, here, we show a clear synergy of robust epithelial-specific ALK5-dependent TGFβ activation with growth-factor signalling that is pro-proliferative and pro-tumourigenic, consistent with a concentration- and cell type-specific response to TGFβ-pathway activation. As such, it is tempting to draw parallels with wound-healing post injury, where robust, albeit transient, activation of TGFβ signalling promotes repair, with the levels of TGFβ activity diminishing as the wound is resolved. The surge in migration and proliferation, during the initial phases of repair, is followed by a TGFβ-induced reduction in the proliferation of the wound-associated epithelium and the engagement of active cell differentiation during wound resolution[20]. By extension, the response of cancer cells to different thresholds of TGFβ signalling would depend both on the levels of circulating TGFβ and their specific mutational landscape. In agreement, we confirm the potent tumour-suppressive effects of TGFβ signalling in *Apc*-mutant cells[22,40], which could help explain why multiple mutations (Wnt- and MAPK-activating) are required to tolerate epithelial TGFβ signalling and switch the cellular response to TGFβ from apoptosis to proliferation. However, the sensitivity of our TGFβ-treated organoid models, which lack the full complement of stromal cell types, might suggest additional niche factors assist the oncogenic switch in response to TGFβ in *Apc.Kras*-mutant cells.

Here, we resolve this apparent paradox and demonstrate that the sustained activation of epithelial-intrinsic—rather than stromal—TGFβ signalling acts as a tumour promoter in the intestinal epithelium, driving gene expression programmes associated with aggressive CRC subtypes, which can be exploited therapeutically. In normal intestinal homoeostasis, concentration gradients of TGFβ/BMP signalling serve to limit the expansion of proliferative progenitor cells that emerge from the intestinal crypt, driving their terminal

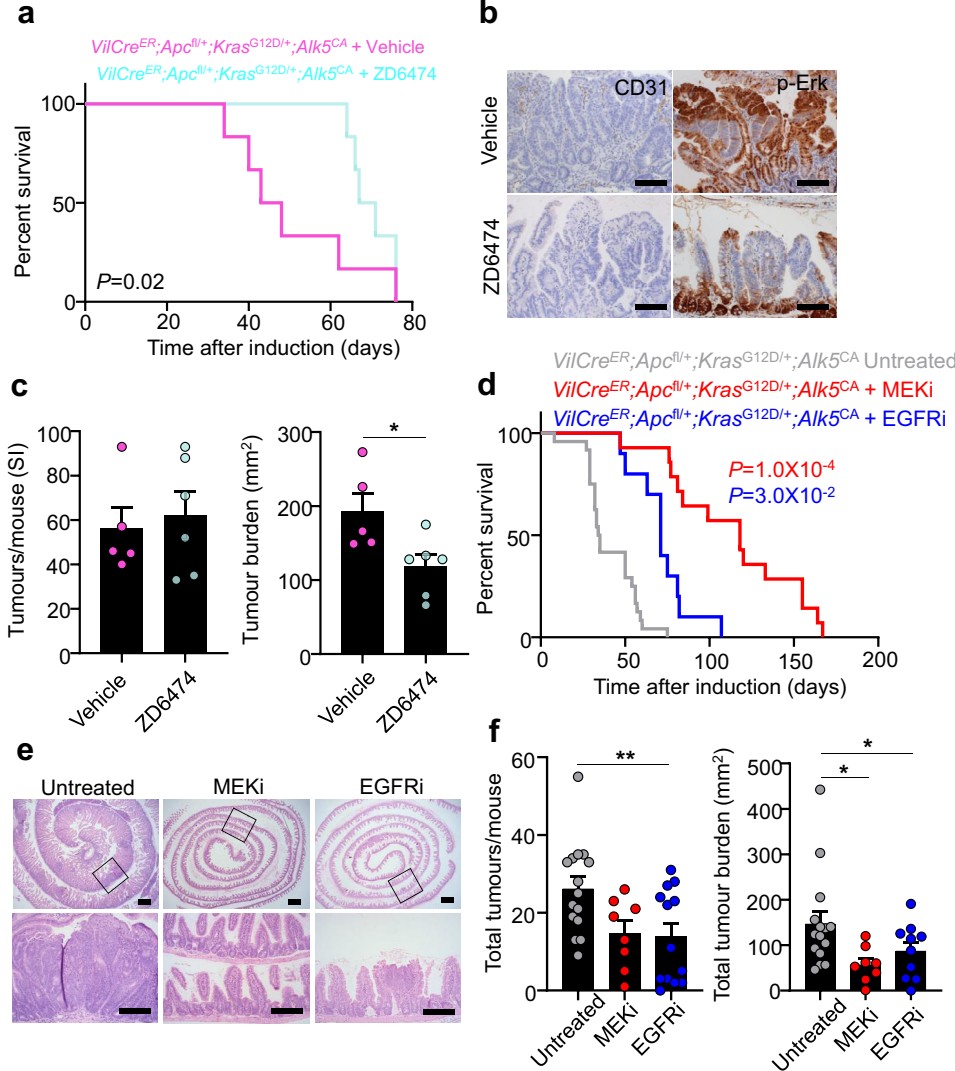

**Fig. 7 | Epithelial TGFβ/ALK5 signalling exposes therapeutic vulnerability to MAPK-targeted therapies. a** Survival plot for *VilCre*[ER];*Apc*[fl/+];*Kras*[G12D/+];*Alk5*[CA] mice treated daily with vehicle or vandetanib/ZD6474 and aged until clinical endpoint following tamoxifen induction. *n* = 6 mice per group. *P* = 0.02, log-rank test. **b** Representative CD31 and p-ERK staining on small intestinal tumour tissue from mice described in **a**. Scale bar, 100 μm. **c** Small intestinal (SI) tumour number (left) and burden (right) per mouse from mice described in **a**. *n* = 5 vehicle (pink), *n* = 6 ZD6474 (aqua) mice. Data were ±s.e.m; *P* = 0.98 (tumour number), *\*P* = 0.02 (tumour burden). Two-tail Mann–Whitney *U*-test. **d** Survival plot for *VilCre*[ER]; *Apc*[fl/+];*Kras*[G12D/+];*Alk5*[CA] mice treated daily with MEK1/2 inhibitor (MEKi) or EGFR inhibitor (EGFRi) and aged until clinical endpoint following tamoxifen induction. *n* = 24 untreated (grey), *n* = 14 MEKi (red), *n* = 10 EGFRi (blue) mice. *P* = 1.0 × 10⁻⁴ (MEKi), *P* = 3.0 × 10⁻² (EGFRi), log-rank test. **e** Representative H&E staining of small intestinal tissue from *VilCre*[ER];*Apc*[fl/+];*Kras*[G12D/+];*Alk5*[CA] mice following indicated treatments. The bottom panels are a magnification of the boxed areas in the corresponding top panels. Scale bar, 100 μm. **f** Total tumour number (left) and burden (right) per mouse from mice described in **d**. *n* = 15 untreated (grey), *n* = 8 MEKi (red), *n* = 13 EGFRi mice (blue). Data were ±s.e.m; Tumour number: *P* = 0.09 (MEKi), *\*\*P* = 0.005 (EGFRi), Tumour burden: *\*P* = 0.01 (MEKi), *\*P* = 0.03 (EGFRi). Two-tail Mann–Whitney *U*-test.

differentiation[37,54,55]. In support, we show that constitutive activation of TGFβ/ALK5 signalling triggers cell death in vitro and in vivo. However, unlike the cytostatic and cytotoxic effects observed in vitro, the intestinal epithelium of *VilCre*[ER];*Alk5*[CA] mice does not collapse and instead responds with a boost in cell proliferation and an expansion of ISCs, which confer short-term functional advantages following epithelial damage, consistent with previous reports implicating TGFβ signalling in the repair and regeneration of the intestinal epithelium post injury[20]. This pronounced mobilisation of the epithelium could be due to compensatory proliferation of surviving epithelial cells[56] or the release of supportive niche factors from stromal populations[57]. Given the plethora of TGFβ ligands and the fact that ALK5 can also mediate GDF signal transduction[58,59], the identity and source of the ligand(s) driving epithelial ALK5 activation in early-stage CRC is currently unclear. Intriguingly, we have previously shown that sustained

TGFβ/ALK5 activation is poorly tolerated in the pancreas and liver[23,24], in contrast to our findings in the intestine, underscoring important tissue-specific consequences of TGFβ/ALK5 activation.

Our pT1 tumour profiling revealed that several signalling cascades that drive an aggressive phenotype are established in early-stage tumours, including an upregulation of all CRIS-related intrinsic signalling compared with less aggressive lesions. In contrast to these tumour-intrinsic results, aggressive pT1 tumours were not associated with the immune and stromal-related signalling that is prognostic for more advanced CRC, namely the CMS1 and CMS4 subtypes, respectively[18,21,53]. Although our findings suggest that stromal predominance is not required for the acquisition of aggressive traits at the earliest stages of CRC initiation, we cannot rule out the possibility that aggressive pT1 tumours will go on to become stromal-rich, if allowed to develop further, and that they represent an initial step in the evolution of aggressive

later-stage disease. This possibility is supported by the finding that a transcriptional signature, derived from tissue taken from the *Alk5*CA-based model at 72 h post induction (in the absence of widespread stromal activation), was sufficient to identify aggressive pT1 human tumours. However, when allowed to develop further in combination with *Apc* and *Kras* mutations, these models go on to upregulate several intrinsic (tumour cell autonomous) and extrinsic (TME-driven) signatures (CRIS-B and CMS4) associated with aggressive later-stage CRC. Nevertheless, given the complexity of human disease, it is unlikely that these models recapitulate the plethora of biological processes that drive early dissemination in human tumours; however, they do lend themselves to the pre-clinical testing and validation of therapeutic vulnerabilities inherent in aggressive early-stage CRC.

TGFβ stimulation has been shown to activate MAPK signalling in a variety of ways[60,61], and co-operate with mutations in the MAPK pathway to promote the pathogenesis of *BRAF-mutant* serrated CRC[22]. Moreover, in concert with Wnt- and MAPK-pathway activation, we show here that sustained TGFβ/ALK5 signalling promotes the initiation and growth of intestinal tumours through the dedifferentiation of terminally differentiated epithelial cells. Mechanistically, the subversion of homoeostatic mitogen gradients, such as RET and EGFR, is sufficient to enhance pro-proliferative Wnt signalling and suppress epithelial shedding, respectively[62,63]. We have previously shown that the activation of NF-κB signalling in the inflamed intestinal epithelium also generates tumour-initiating cells from differentiated lineages[64]. As such, future studies should investigate whether sustained TGFβ/ALK5 signalling acts upstream, downstream, or in parallel with the pro-inflammatory NF-κB pathway to promote tumour-initiating capacity.

The efficacy of targeted therapies in cancer is often undermined by the development of resistance, which can occur in a variety of ways[65,61]. Importantly, here, we have identified a promising therapeutic vulnerability conferred by TGFβ/ALK5 signalling, namely the preferential upregulation of specific growth-factor signalling modalities that can be readily targeted to extend survival. Of note, the substantial survival benefit and suppression of tumour growth, observed in sapitinib-treated *VilCre*ER*;Apc*fl/+*;Kras*G12D/+*;Alk5*CA mice, is promising, particularly as the potency of EGFR-targeted therapies rapidly diminishes following *Kras* mutation[46]. Whilst sapitinib might not show efficacy as a single agent in established human tumours, it has the potential to be rationally combined with agents selected based, in part, on the molecular mechanisms driving innate and acquired resistance to monotherapy. The recent findings from the BEACON clinical trial, wherein doublet and triplet combinations of EGFR/MAPK-targeted drugs achieved modest efficacy in *BRAF*-mutant metastatic CRC[66], and the realisation that mutant-specific KRAS-G12C inhibitors cannot achieve therapeutic benefit alone[67], hold promise that analogous combination regimens could be of efficacy in tumours with high epithelial TGFβ (e.g. CRIS-B). Indeed, the striking impact of MEK inhibition in *VilCre*ER*;Apc*fl/+*;Kras*G12D/+*;Alk5*CA mice and organoid models suggests that new, potentially more effective treatment combinations could be found to overcome resistance in emergent TGFβ-high *Kras*-mutant CRC.

Given the current lack of understanding of the biology underpinning early-stage aggressive lesions in CRC, the data presented here illuminate transcriptional biomarkers to stratify lethal from non-lethal early-stage CRC and provide compelling functional evidence that the aberrant activation of epithelial TGFβ signalling cooperates with other common CRC mutations to promote tumour formation. Together, these findings highlight potential therapeutic options for the increasing numbers of CRC patients diagnosed with early-stage CRC during screening.

## Methods

### pT1 patient cohort

The pT1 samples were collected via the Northern Ireland Biobank (study reference: NIB16-0215), which has approval from ORECNI

(16/NI/0030) to collect, store and distribute samples to researchers[68]. Written informed consent from patients was obtained prior to acquisition of the tissue sample for research use. Gene expression profiles for pT1 cohorts were generated using Almac Diagnostics Xcel Array platform as part of the S:CORT consortium[69]. The annotation file 'Xcel Annotations, CSV format, Release 36', (http://www.affymetrix.com/support/technical/byproduct.affx?product=xcel) was downloaded from Affymetrix. Raw microarray.CEL file data were normalised using Robust Multi-array Average (RMA) in Partek Genomics Suite software (v6.6; Partek Inc.). Principal components analysis (PCA) was used to assess the quality of the data. Prior to analysis, data were collapsed to the probeset with the highest mean value for each gene using the 'collapseRows' function from the WGCNA R package (v1.68). Differential gene expression analysis was performed using limma (v3.40.6) R package. pT1 gene expression dataset and clinicopathological information are provided from S:CORT, with transcriptional data available on GEO under accession number GSE162667. Mutation data were generated by DNA target capture (SureSelect, Agilent) spanning all coding exons of 80 CRC driver genes followed by next generation sequencing using the Illumina HiSeq2000 platform at the Sanger Institute. Variant calling was performed with Caveman for point mutations and Pindel for indel mutations. Mutations were visualised using the ggplot2 (v3.3.5) R package.

### FOCUS cohort

Through the S:CORT consortium, we also assessed tumour specimens and associated clinical data a subset of the FOCUS clinical trial cohort, that is a large UK-based randomised controlled trial comparing different strategies of sequential or combination therapies of 5FUFA with or without oxaliplatin or irinotecan as first- or second-line therapies in patients with newly diagnosed advanced colorectal cancer[70]. The FOCUS clinical trial was performed in accordance with the Declaration of Helsinki. All subjects provided written informed consent for further research on their samples at the time of consent to the clinical trial. Both the original clinical trial (FOCUS Ref: 79877428) and the data reported here (S:CORT ref 15/EE/0241) were approved by the National Research Ethics Service in the United Kingdom. Transcriptional data of 361 samples used in this study is available on GEO under accession number GSE156915.

### Memorial Sloan Kettering stage II CRC IHC cohort

The CRC IHC cohort was comprised of 283 stage II patients and was previously described[71]. For tissue microarrays (TMAs), three separate 2-mm tissue cores each from tumour tissue and matched normal colon mucosa were removed from each donor paraffin block and transferred to tissue array blocks using a robotic TMA arrayer (TMA Grand Master, 3DHistech). The study protocol was reviewed and approved by the International Review Board of Memorial Sloan Kettering Cancer Center. Data were acquired retrospectively and in an anonymized manner such that patient consent was not required as determined by the IRB.

### Multiplexed immunofluorescence

Multiplexed immunofluorescence (MxIF) iterative staining of the stage II CRC TMAs was performed using Cell DIVE™ technology (Leica Microsystems (formerly GE Healthcare), Issaquah, WA)[47]. This technique involves iterative staining and imaging of the same sample with 60+ antibodies on the same tissue section. For the purpose of this study, 5 markers were down-selected for analysis: p-mTOR (Ser 2448) clone 49F9 (#2976; Cell Signaling), TGF1-beta clone TB21 (#05-1423; Millipore Sigma), p-S6 (Ser 235/236) clone D57.2.2E (#4858; Cell Signaling) and 4EBP1 clone 53H11 (#9644; Cell Signaling). Antibodies for cell and subcellular compartment segmentation included: NaKATPase clone EP1845Y (#2047-1; Epitomics), ribosomal S6 (#2217; Cell Signaling), pan-cytokeratin clone PCK-26 (#C1801; Sigma) and DAPI (4′,6-diamidino-2-phenylindole) stain. All nuclei in the image

(stromal, immune and epithelial) were segmented using DAPI signal. Cells in the epithelial compartment were further segmented using pan-cytokeratin, p-S6, and NaKATPase stains, as previously described[47]. An epithelial region mask was also generated using combined cytokeratin, NaKATPase and S6 images and this was used to separate epithelial from stromal cells. Images and cell data underwent a multistep quality control review process as previously described[72], which involved visual assessment of staining patterns compared to controls and/or previous findings with these antibodies, followed by visual inspection of cellular segmentation. After the QC steps, the data underwent exposure time correction, log2 transformation to handle the skewness of the marker intensities, and normalisation to remove any slide batch effects. One patient was excluded due to an inadequate number of tumour cells for analysis thereby reducing the stage II CRC cohort size to 282.

## MxIF data analysis
After the pre-processing steps, unsupervised k-means clustering with k = 2–15 was applied on the intensity of the all 15 markers (key components of mTOR and TGFβ pathways) for epithelial cells: p-S6, S6, β-catenin, ALDH1, 4EBP1, p-4EBP1, ERK, p-ERK, TGFB1, EGFR, p53, p-mTOR, Wnt5, p38MAPK and EZH2. Extreme values (less than 1%-tile and greater than 99%-tile) were capped, and marker intensities were standardised to zero mean and single standard deviation. Consensus clustering[73] and proportional of ambiguously clustered (PAC)[74] were used to determine the optimal number of groups that consistently separated the cells. Following cluster analysis, the proportion of cells in each cluster, was determined for each patient (patient cluster profile). The median staining intensities of TGFB1, p-mTOR and p-S6 and 4EBP1 markers at single cell level was used to visualise the association of each cluster of stage II CRC patients with pT1 signature.

## Patient survival analysis
Kaplan-Meier survival analysis was performed using the function survfit() of the R package 'survival' (v3.2-13). Stage II CRC patients were ranked according to the proportion of single cells associated with cluster 4. Patients with more than 50% of their epithelial cells associated with cluster 4 were classified as 'high' (n = 42) and those with fewer than 50% of their epithelial cells associated with cluster 4 were classified as 'low' (n = 240) (Supplementary Table 2). Differences in relapse-free survival (RFS) were compared using the log-rank test. Cox proportional hazards regression was performed using the function coxph() from the R package 'survivalAnalysis' (v0.2.0) to calculate hazard ratios (HR) and 95% confidence intervals (95% CI). Survival curves were generated with ggsurvplot from the R package 'survminer' (v0.4.9).

## Quantification of immune cell and fibroblast content
Transcriptome-based stromal and immune scores were generated using the 'MCPcounter.estimate' function in MCPcounter R package (v1.2.0) and compared to the digital pathology-based stromal scores calculated using QuPath. Correlation analyses were performed using the 'cor.test' function from the R stats package, with the 'Spearman' method for the T1 cohort and with the 'Pearson' method for the FOCUS cohort. The default setting was used for all other parameters. These statistical analyses were performed using stats package R (v4.1.2) and RStudio (v1.1.447).

## Classification of of the pT1 cohort using the CMS and CRIS classifiers
The consensus molecular subtype (CMS) of each patient tumour was determined using CMS classifier (v1.0.0); the 'classifyCMS.RF' function R package. CRC intrinsic subtype (CRIS) classifications[21] were predicted using the Nearest Template Prediction (NTP) module (v4) on GenePattern (https://genepattern.broadinstitute.org/). To account for the small sample sizes, we also utilised the CRIS and CMS classifiers in conjunction with gene set enrichment analysis (GSEA) and single-sample GSEA (ssGSEA) to determine the degree of enrichment of the classifier gene sets in individual pT1 patient samples.

## Gene set enrichment analysis
A fast implementation of pre-ranked Gene Set Enrichment Analysis (FGSEA) using the fgsea (v1.18.0) R package was performed on log expression ratio of relapse vs non-relapse cases to assess enrichment of CMS/CRIS subtypes and also enrichment of CAF and stromal gene sets[21] and Hallmark gene set of MSigDB[28] resource. R package msigdbr (v7.4.1) was used to retrieve the Hallmark gene set. Benjamini–Hochberg FDR <0.2 considered as significant.

## Digital histology
Digital image analysis of whole slide images was conducted in QuPath 1.2.0. Manual tissue detection was guided by the pathologist annotation. DoG superpixel creator (Downsample factor 8, Gaussian sigma 5 μm, Minimum intensity threshold 5, Maximum intensity threshold 230, Noise threshold 1) segmented tissue for further analysis. Coherence features (Magnification 5, Stains: H&E, Tile diameter 25 μm, include basic statistics) were added and intensity features (preferred tile size 2 μm, region: ROI, tile diameter 25 μm, colour transformations to include; Optical density sum, Hematoxylin, Eosin, Residual. Basic features to include; Mean, Standard deviation, Min&Max, Median. Compute Haralick features with a Haralick distance of 1 and 32.0 bins) computed to enable a detection classifier (Random Trees using single threshold) to be created. Annotated regions of tumour epithelium and stroma, on numerous cases with varying histology were used to train the classifier (Build and Apply).

## Mouse studies
All animal experiments were performed in accordance with UK Home Office regulations (under project licence 70/8646) and by adhering to the ARRIVE guidelines with approval from the local Animal Welfare and Ethical Review Board of the University of Glasgow. Mice were housed in conventional cages in an animal room at a controlled temperature (19–23 °C) and humidity (55 ± 10%) under a 12 h light/dark cycle and were fed standard rodent chow and water ad libitum. All mice were maintained on a mixed C57BL/6 background. Mice of both sexes, aged 6–12 weeks, were induced with a single intraperitoneal (i.p) injection of 2 mg tamoxifen as indicated (Sigma-Aldrich, #T5648). The transgenes/alleles used for this study were as follows: $VilCre^{ER32}$, $Apc^{580S42}$, $Kras^{G12D44}$, $Alk5^{CA33}$, $Alk5^{fl75}$, $Bcat^{Ex3\ 43}$ and $Smad4^{fl38}$. For tumour growth studies, mice were aged until they showed clinical signs of intestinal disease (anaemia, hunching and/or weight loss). For treatments, mice were dosed one day after tamoxifen injection using the following regimens: ALK5-inhibitor/ (ALK5i)/AZ12601011 (50 mg/kg, twice-daily oral gavage, AstraZeneca), vandetinib/ ZD6474 (50 mg/kg, once-daily oral gavage, LC Laboratories; #V-9402), selumetinib/AZD6244 (25 mg/kg, twice-daily oral gavage, AstraZeneca; AZ12252244), sapitinib/AZD8931 (25 mg/kg, twice-daily oral gavage; AstraZeneca). All of the listed drugs/reagents were made up in 0.5% hydroxypropyl methylcellulose (HPMC) and 0.1% Tween-80. For regeneration studies mice were exposed to whole-body irradiation with γ-rays (10 Gy).

## Organoid culture
Small intestinal crypts were isolated from individual mice of the indicated genotypes and expanded into organoids. Briefly, 10 cm of proximal small intestine was removed, flushed with water, cut longitudinally and cut into small (1–2 cm) pieces. Tissue pieces were incubated in chelation buffer (5 mM EDTA in PBS) for 30 min on roller at 4 °C. Gut tissue was washed briefly in PBS and shaken to release crypts from stroma. Crypts were filtered through 70 μm cell strainer and collected in 50 ml tube, washed in cold PBS and

centrifuged for 5 min at $300 \times g$. Crypts were resuspended in Matrigel (BD Bioscience, #354234) and plated in pre-warmed tissue culture plates. Cells were overlaid with growth medium comprising Advanced DMEM/F12 (#12634010) supplemented with penicillin-streptomycin (#15070063), 10 mM HEPES (#15630056), 2 mM glutamine (#25030081), N2 (#17502048), B27 (# 17504044) (all from Gibco, Life Technologies), 100 ng/ml Noggin (#250-38), 500 ng/ml R-Spondin (#315-32) and 50 ng/ml EGF (#315-09) (all from PeproTech) following Matrigel polymerisation. Each organoid line was independently derived from a different donor mouse. R-Spondin was not included in the culture medium of *Apc*-mutant organoid cultures. To induce Cre-mediated recombination in vitro, 5 μM 4-hydoxytamoxifen (Tmx; Merck, #T176) was added to the culture medium. ALK5-inhibition was achieved in vitro by adding ALK5i (AstraZeneca, AZ12601011) to the culture medium at a final concentration of 10 nM/ml. Recombinant TGFβ1 (5 ng/ml; Peprotech, #100-21) was mixed with standard growth medium and replenished every other day.

### Organoid viability
Organoid viability was measured as previously described[76,77]. Briefly, organoids were mechanically disrupted and resuspended in TrypLE™ Express (Thermo Fisher Scientific, #12604013) with 100–200 U DNAse (Sigma, #4716728001) for 1 h at 37 °C. Cells were passed through a 40-μm strainer, counted, and seeded in 200 μl Matrigel/PBS (1:1 mixture) in a 24-well plate and overlaid with standard growth medium plus treatments 48 h after seeding (described above). Viability was quantified using CellTiter-Blue® (Promega, #G8080) at indicated timepoints, in combination with morphological grading/scoring of organoids with clear central lumen. A total of 50-100 organoids from 3 representative images were analysed per condition in $n = 2$ independent experiments.

### Mouse RNA isolation and qPCR
RNA isolation was performed using the RNeasy mini kit (Qiagen, #74104) and cDNA was synthesised using the high capacity cDNA reverse transcription kit (Thermo Fisher Scientific), #4368813). SYBR Green (Thermo Fisher Scientific, #F410XL) qPCR reactions were performed with the BioRad CFX96 Real-Time System PCR system using BioRad software using CFX Maestro Software 2.3. Relative fold change in gene expression was calculated using the $2^{-\Delta\Delta Ct}$ method. All $\Delta\Delta$Ct values were normalised to the housekeeping gene *Gapdh*. The sequences of the primers used for qPCR were as follows: *Gapdh* forward, 5′-GAAGGCCGGGGCCCACTTGA-3′; *Gapdh* reverse, 5′-CTGGG TGGCAGTGATGGCATGG-3′; *Alk5$^{CA}$* forward, 5′-TTGTGAACAGAA GTTAAGGC-3′; *Alk5$^{CA}$* reverse, 5′-AGCATAATCAGGAACATCAT-3′; *Alk5* forward, 5′-TCTGCATTGCACTTATGCTGA-3′; *Alk5* reverse, 5′-AAAGGG CGATCTAGTGATGGA-3′; *Smad7* forward, 5′-GGCCGGATCTCAGGCA TTC-3′; *Smad7* reverse, 5′-TTGGGTATCTGGAGTAAGGAGG-3′; *SerpA1* forward, 5′-CCCCACGGAGATGGTTATAG-3′; *SerpA1* reverse, 5′-ATCAC TTGCCCCATGAAGAG-3′; *Puma* forward, 5′-AGCAGCACTTAGAGTC GCC-3′; *Puma* reverse, 5′-CCTGGGTAAGGGGAGGAGT-3′; *Noxa* forward, 5′-GCAGAGCTACCACCTGAGTTC-3′; *Noxa* reverse, 5′-CTTTTG CGACTTCCCAGGCA-3′; *Bax* forward, 5′-TGAAGACAGGGGCCTTTTTG-3′; *Bax* reverse, 5′-AATTCGCCGGAGACACTCG-3′.

### Mouse RNA sequencing and analysis
Total RNA was extracted from whole small intestinal tissue. RNA integrity was analysed with a NanoChip (Agilent Technologies RNA 6000 Nanokit #5067-1511). A total of 2 μg of RNA was purified via poly(A) selection. The libraries were run on an Illumina NextSeq 500 sequencing system using the High-Output kit v2.5 (75 cycles; Illumina #20024906). GSEA analysis was performed using the fgsea (v1.10.1) R package on shrunken log expression ratio of *VilCre$^{ER}$;Apc$^{fl/fl}$;Kras$^{G12D/+}$;Alk5$^{CA}$* (AKA$^{CA}$)

vs WT models. The comparison gene sets were obtained from refs. [22,31]. A murine CRIS-B gene set was created from the human genes associated with CRIS-B from the original study which were converted into their mouse orthologues using data obtained from the HUGO Gene Nomenclature Committee (HGNC) and Mouse Genome Informatics (MGI) databases. Single-sample gene set enrichment analysis (ssGSEA) was run, using the ssGSEA method in GSVA R package (v1.32.0) with the murine CRIS-B consisting of 63 genes to assign CRIS-B scores to normalised count RNA-Seq data from the various mouse models which had a pseudocount of 1 and been log2 transformed. The R package pamr (v1.55) was used to generate a classifier from transcriptional data from *VilCre$^{ER}$;Alk5$^{CA}$* and *VilCre$^{ER}$;Alk5$^{+/+}$* mice. Using 3-fold cross-validation and a threshold of 4.98 (misclassification error = 0), a classifier containing 60 genes was created. To allow this classifier to be applied to human samples, the mouse genes were converted into their human orthologues using the R package biomaRt (v2.40.5). Using the CMScaller package (v0.99.1), the NTP method was then applied to the human gene template (54 genes) to classify the samples in the pT1 cohort into WT-like ($n = 17$) and ALK5$^{CA}$-like ($n = 10$).

### RNA in situ hybridisation
In situ hybridisation using probes for *Olfm4* (#311838), *Itgβ6* (#312508), *Alk5* (#406208), *Alk5$^{CA}$* (#431048), *Smad7* (#429418) and *Ret* (#431798) mRNA (all from Advanced Cell Diagnostics) was performed using RNAscope 2.5 LS Reagent Kit–BROWN (Advanced Cell Diagnostics) on a BOND RX autostainer (Leica) according to the manufacturer's instructions.

### Mouse immunohistochemistry and immunofluorescence
Intestines were flushed with water, cut open longitudinally, pinned out onto silicone plates and fixed in 10% neutral buffered formalin overnight at 4 °C. Fixed tissue was rolled from proximal to distal end into swiss-rolls and processed for paraffin embedding. Tissue blocks were cut into 5 μm sections and stained with haematoxylin and eosin (H&E). Immunohistochemistry (IHC) and immunofluorescence (IF) were performed on formalin-fixed intestinal sections according to standard staining protocols. Primary antibodies used for IHC and IF were against: p-SMAD3 (1:50, Abcam, #ab52903), β-catenin (1:100; BD Transduction Labs, #610154), lysozyme (1:300; Agilent Technologies, #A099), BrdU (1:250, BD Biosciences, #347580), CD3 (1:100, Abcam, #ab16669), Ly6G (1:60,000, BioxCell, #BE0075-1), F4/80 (1:100, Abcam, #ab6640), cleaved caspase-3 (1:500, Cell Signaling Technology, #9661), CD31 (1:75, Abcam, #ab28364) and p-ERK (1:400; Cell Signaling Technology, #9101). Representative images are shown for each staining. For image analysis Image J, version 1.52 h and Perkin Elmer Harmony high-content and imaging analysis software, version 4.8 were used.

### Statistical analyses
The smallest sample size that could give a significant difference was chosen in accordance with the 3Rs. Given the robust phenotype of the *Apc$^{fl/fl}$* mice and assuming no overlap in the control versus the experimental group, the minimum sample size was 3 animals per group. For analysis of organoid cultures, investigators were blinded when possible. Investigators performed all histological quantification blinded to genotype or treatment. Statistical analysis was performed with GraphPad Prism v8.0.0 for Windows (GraphPad Software) using one-tailed Mann–Whitney *U*-tests unless otherwise stated. Statistical comparisons of survival data were performed using the log-rank (Mantel-Cox) test. For individual value plots, data are displayed as mean ± standard error of the mean (s.e.m.). $P$ values < 0.05 were considered significant. Statistical tests and corresponding $P$ values are indicated in the figure legends and figures, respectively.

## Reporting summary

Further information on research design is available in the Nature Portfolio Reporting Summary linked to this article.

## Data availability

Raw and processed RNA sequencing data that support the findings in this study and available on GEO under accession number GSE182019. Large raw image/micrograph files that support the findings in this study are not provided with the source data and are available from the corresponding authors upon request. pT1 clinicopathological information are provided in Supplementary Table 1, and raw and processed transcriptional data is available on GEO under accession number GSE162667. Raw and processed transcriptional data from FOCUS cohort data are available on GEO under accession number GSE156915. The CRC cohort was comprised of 282 stage II patients and was previously described in Oncotarget. 2020 Feb 25; 11(8): 813–824; https://doi.org/10.18632/oncotarget.27491. Source data are provided with this paper.

## Code availability

Custom codes and scripts are available on https://github.com/MolecularPathologyLab/T1-Alk5ca.

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

## Acknowledgements

We thank the Core Services and Advanced Technologies at the Cancer Research UK Beatson Institute (C596/A17196 and A31287), and particularly the Biological Services Unit, Histology Service and Molecular Technologies. We are thankful to members of the Sansom and Dunne labs for discussions of the data and manuscript. O.J.S. and his lab members were supported by Cancer Research UK (A28223, A21139, A12481, A17196 and A31287). D.J.F and J.D.G.L were supported by UK Medical Research Council (MR/R017247/1 and MR/NO21800/1 respectively). P.D.D and the Belfast team were supported by Cancer Research UK (ACRCelerate award; A26825 and Early Detection award; A29834), a joint Cancer Research UK and UK Medical Research Council programme (S:CORT MR/M016587/1), the NI HSC R&D Experimental Cancer Medicine Centre (ECMC) programme (COM/5535/19) and the CRUK Infrastructure Oversight Committee (25176) for Belfast ECMC. We thank support from an MRC NMGC Cancer Cluster grant (grant no. MC-PC_21042). Additionally, M.L. was supported by funding to the Health

Data Research Wales Northern Ireland Substantive site and to DATA-CAN, the UK's Health Data Research Hub for Cancer. F.G. and S.C. were supported by National Cancer Institute of the National Institutes of Health under award number R01CA208179 and the NIH US-Ireland R01 award to Northern Ireland Partners supported by HSCNI (STL/5715/15). The pT1 samples used in this research were received from the Northern Ireland Biobank, which has received funds from the HSC Research and Development Division of the Public Health Agency in Northern Ireland and the Friends of the Cancer Centre. CRC samples for multiplexed analysis were provided by Dr. Jinru Shia. Collection of clinical data was facilitated by the Northern Ireland Cancer Registry, which is funded by the Public Health Agency, Northern Ireland. Figure 3c and Supplementary Figs. 1a, 3a and 5g were created with BioRender.com.

## Author contributions

D.J.F, D.F.V, R.A, P.D.D. and O.J.S. designed the research and interpreted the data. D.J.F, D.F.V, P.G, P.C, H.L.D. and R.A.R. performed experiments and analysed data. D.J.F, N.G. and J.L. performed and analysed organoid experiments. H.G.C., M.B.L. and P.D.D. identified pT1 patient data and samples. N.C.F., M.B.L. and P.D.D. performed histological assessments. R.A., F.G., S.C. and P.D.D. performed analysis on multiplex cohort. A.J.M.., K.L.R., R.A., A.M.B.M., K.G., E.K.M., W.C., A.H. and E.M. performed transcriptomics and analysed transcriptomic data. J.D.G.L., R.J., D.S., L.B., K.L.R., E.M.K., D.B.L., F.G., S.C., H.G.C., M.B.L., A.B., T.S.M., A.D.C., M.L., S.J.L., S.T.B., G.J.I. and J.v.R. provided reagents and advice. D.J.F. and C.N. performed IHC and ISH. D.J.F., R.A., P.D.D. and O.J.S. wrote the paper. All authors contributed to the final version of the paper.

## Competing interests

M.L. has received an unrestricted educational grant from Pfizer, for research unrelated to this work. M.L. has received honoraria from Pfizer, EMD Serono, Roche and Carnall Farrar unrelated to this work. The remaining authors declare no competing interests.

## Additional information

**Dustin J. Flanagan** [1,2,3,17] ✉, **Raheleh Amirkhah** [4,17], **David F. Vincent** [1,17], **Nuray Gunduz** [1], **Pauline Gentaz** [1], **Patrizia Cammareri** [1], **Aoife J. McCooey** [4], **Amy M. B. McCorry** [4], **Natalie C. Fisher** [4], **Hayley L. Davis** [5], **Rachel A. Ridgway** [1], **Jeroen Lohuis** [6], **Joshua D. G. Leach** [1,7], **Rene Jackstadt** [1,8], **Kathryn Gilroy** [1], **Elisa Mariella** [9], **Colin Nixon** [1], **William Clark** [1], **Ann Hedley** [1,10], **Elke K. Markert** [1,7], **Douglas Strathdee** [1], **Laurent Bartholin** [11], **Keara L. Redmond** [4], **Emma M. Kerr** [4], **Daniel B. Longley** [4], **Fiona Ginty** [12], **Sanghee Cho** [12], **Helen G. Coleman** [4,13], **Maurice B. Loughrey** [4,13,14], **Alberto Bardelli** [9], **Timothy S. Maughan** [15], **Andrew D. Campbell** [1], **Mark Lawler** [4], **Simon J. Leedham** [5], **Simon T. Barry** [16], **Gareth J. Inman** [1,7], **Jacco van Rheenen** [6], **Philip D. Dunne** [1,4,17] & **Owen J. Sansom** [1,7,17] ✉

[1]Cancer Research UK Beatson Institute, Glasgow, UK. [2]Department of Biochemistry and Molecular Biology, Monash University, Melbourne, Australia. [3]Cancer Program, Biomedicine Discovery Institute, Monash University, Melbourne, Australia. [4]The Patrick G. Johnston Centre for Cancer Research, Queen's University Belfast, Belfast, UK. [5]Wellcome Trust Centre for Human Genetics, University of Oxford, Oxford, UK. [6]Department of Molecular Pathology, Oncode Institute, The Netherlands Cancer Institute, Amsterdam, The Netherlands. [7]Institute of Cancer Sciences, University of Glasgow, Glasgow, UK. [8]Heidelberg Institute for Stem Cell Technology and Experimental Medicine (HI-STEM gGmbH) and Deutsches Krebsforschungszentrum (DKFZ), Heidelberg, Germany. [9]Department of Oncology, University of Torino, Candiolo Torino, Italy. [10]University of Newcastle upon Tyne, Newcastle, UK. [11]INSERM Centre de Recherche en Cancérologie de Lyon, Lyon, France. [12]GE Global Research Center, Niskayuna, NY, USA. [13]Centre for Public Health, Queen's University Belfast, Belfast, UK. [14]Department of Cellular Pathology, Belfast Health and Social Care Trust, Belfast, UK. [15]CRUK/MRC Oxford Institute for Radiation Oncology, University of Oxford, Oxford, UK. [16]Bioscience, Oncology R&D, AstraZeneca, Cambridge, UK. [17]These authors contributed equally: Dustin J. Flanagan, Raheleh Amirkhah, David F. Vincent, Philip D. Dunne, Owen J. Sansom. ✉e-mail: dustin.flanagan@monash.edu; o.sansom@beatson.gla.ac.uk

