## [Peer Review File · Nature Communications]

REVIEWER COMMENTS

Reviewer #1 (Remarks to the Author): Expert in colorectal cancer functional genomics and models

In this manuscript by Flanagan et al., the authors report their analysis on the paradox of TGF β in CRC mostly in early tumorigenesis. They demonstrate that activation of TGF β in the epithelium can act as a tumour promoter when combined with the proper gene mutations leading to a perfect storm in early diagnosed tumours. Although, the manuscript is sometime challenging to read, it shows interesting and exciting results. Most experiments shown here support the conclusion drawn by the authors. I have few comments, but they should be resolved as they would elevate a very good manuscript to publication standards in Nature Communications.

Comments:

1. The authors showed as a target gene for TGF β , Smad7 (Figure 2, extended Figure 3, extended Figure 6). In addition, they have also generated a compound mouse with their model with a Smad4 floxed mouse to demonstrate a rescue of their phenotype (Figure 2 and extended Figure 2). Smad7 is the iSmad that will affect all TGF β superfamily members and more precisely TGF β and BMP (unlike Smad6 that is specific to the BMP pathway). The same apply for Smad4 which is the co-Smad. Smad4 form an heterotetrametric complex with the R-Smads (Smad 2/3- TGF β ; Smad1/5/8-BMP) then allowing translocation of the complex into the nucleus to affect target genes. As Smad4 and Smad7 are seen at the interface of both BMP and TGF β signaling pathway, the authors must demonstrate that the BMP signaling pathway is not affected in their model. Especially since BMP signaling plays such an important role in stem cell regulation in the GI tract and any deregulation in the BMP/WNT axis could have an impact by itself in some of the phenotype observed by the authors.

2. Figure 4b and b; In light of these results, the significant increase come from mostly 2 individuals from the 17 mutants analyzed compared to 12 controls. To this reviewer, this does not represent a clear shift in the phenotype as suggested by the authors. Thus, the biological significance of this results is overinterpreted and need to be readdressed accordingly.

3. Reading of the manuscript is sometime very challenging and difficult to follow. I understand that the authors have numerous results to show and I commend them for that, but sometime the asynchronous sequence in the figures complexify the reading and breaks the flow for the reader. As for example there is a sequence in the paper where the reader has to go from Figure 4; extended Figure 5; extended Figure 6; Figure 6; extended Figure 6.

Reviewer #2 (Remarks to the Author): Expert in gastrointestinal cancer organoids and functional genomics

To date, the contribution of TGFb signaling to malignancy in CRC is controversial. Flanagan et al. use engineered mouse models and murine intestinal organoids to study the effect of constitutive active TGFb-signalling in the intestinal and colon epithelium (by conditionally expressing a mutant TGFbR1 that leads to ligand independent dimerization and activation of TGFb signaling). First, they show that active TGFb-signalling in the wildtype intestinal epithelium leads to apoptosis in vitro in organoids, but to enhanced cell proliferation in vivo. Second, they show that in an Apc mutant background constitutive TGFb activation leads to apoptosis in intestinal epithelial cells in vitro and in vivo. Third, they found that if in addition to the APC mutation intestinal epithelial cells also contain a Kras-G12D mutation they are protected from TGFb mediated apoptosis. In addition, they demonstrate that in this background TGFb expression leads to increased malignancy. Interestingly, they also found that the gene expression signature of these mice correlates with the signature of a highly aggressive T1 subtype of CRC in patients, and they demonstrate that inhibition of MEK and EGFR counteracts the effect of overactive TGFb-signalling in tApc-KrasG12D mice.

I recommend this manuscript for publication in Nature Communications. The experimental design of this study is coherent and the work adds substantial evidence to the controversy of TGFb-signaling in CRC. In addition, it is a valuable information that MEK inhibition could be beneficial in tumors with overactive TGFb signaling. However, resolving the following comments may add to the quality of the manuscript.

Main points

- The work heavily relies on findings derived by autochthonous mouse models and corresponding murine organoid lines. Proofing key aspects of the proposed mechanisms in human cells would allow to generalize the conclusions of this study to human cancer. This may be achieved by testing TGFb response in human intestinal organoids that contain LOF mutations in APC (using CRISPR) and express KrasG12D (using a lentiviral vectors from addgene).
- For some of the bar graphs that quantify histology results no example images are shown (for example Figure 2e,f). Here it would be interesting to see in which regions of the epithelium cells are apoptotic or proliferating.
- Figure 2: It is somehow surprising that in this experimental setup constitutive active TGFb did not perturb the stem cell niche. Were mice also followed up mice for longer periods than 1 week after Tmx treatment?

- Figure 3: Was the effect of TGF-beta activation also assessed in other Wnt activating backgrounds such as Axin2 knock out? This could be done by genetically disrupting Axin2 using CRISPR in murine intestinal VilCreER x AlkCA organoids, and testing whether this line behaves similar as Apc-mutant VilCreER x AlkCA organoids.

- Extended Fig 6a: How often were organoids passaged in TGFb medium? In the brightfield image some organoids appear apoptotic. It would be important to show viability after passaging. In addition, it seems as in the study of Drost et al. human small intestinal organoids mutant for KrasG12D and Apc cannot grow without Noggin (see publication Drost et al. Nature 2017). Thus, it would be interesting to also perform experiments in human organoids in this study (see point 1). If the outcome in human organoids is not identical to murine organoids I would still recommend publication of the manuscript, but this information would be valuable for the field.

Minor points

- Do the authors have a hypothesis what could protect the intestinal epithelium in vivo of strong cell death induction compared to epithelial-only organoid cultures? Are there factors secreted from the stroma/mesenchyme that could mediate this effect?

- Lines 158-160: A statistical analysis would help to interpret the data, even though I acknowledge that the sample size is relatively small and the derived p-values should be interpreted with caution.

Reviewer #3 (Remarks to the Author): Expert in colorectal cancer genomics, in vivo models, systems biology, and therapy

This manuscript describes exciting research to characterise the role of TGFb in “born-to-be bad” early-stage colorectal cancers (CRC). Interestingly, TGFb is associated with stroma in late-stage CRC, whereas the same TGFb is related to the epithelium of the early-stage CRC. The study includes a range of in vivo models that addresses the role of TGFb in the presence of APC/KRAS aberrations. Following are a few comments:

- Extended Fig1 shows PCA with separation of relapse and non-relapse samples by PC2. It would be good to show the clustering by k-means or other methods within the PCA plot.
- It is unclear what the significance of Figures 1a and b. Shouldn't these be supplementary figures?
- It would be helpful to include Ext. Fig 1C as main figures.
- "CMS4 and CRIS-B have previously been associated with poor prognosis in later stage disease". The authors need to check this statement for validity and cite them accordingly.
- References to certain statements from literature are missing.
- Where comments about correlation have been made (significantly higher or lower), P values and statistical tests applied are not always stated.
- It would be great to highlight the potential translational aspect of the findings.

Referee expertise:

Referee #1: Colorectal cancer functional genomics and models

Referee #2: Gastrointestinal cancer organoids and functional genomics

Referee #3: Colorectal cancer genomics, in vivo models, systems biology, and therapy

We are grateful to the referees for their time spent reviewing our manuscript and for their mostly positive comments and constructive criticisms. We have taken on board the referees' comments and revised our manuscript accordingly. Below, we have provided a point-by-point response to the referees' comments, with the referees' comments in black print and our responses highlighted in blue. In addition, we have included a Figure especially for the review process.

Reviewer #1 (Remarks to the Author):

In this manuscript by Flanagan et al., the authors report their analysis on the paradox of TGF β in CRC mostly in early tumorigenesis. They demonstrate that activation of TGF β in the epithelium can act as a tumour promoter when combined with the proper gene mutations leading to a perfect storm in early diagnosed tumours. Although, the manuscript is sometime challenging to read, it shows interesting and exciting results. Most experiments shown here support the conclusion drawn by the authors. I have few comments, but they should be resolved as they would elevate a very good manuscript to publication standards in Nature Communications.

Comments:

1. The authors showed *Smad7* as a target gene for TGF β , *Smad7* (Figure 2, extended Figure 3, extended Figure 6). In addition, they have also generated a compound mouse with their model with a *Smad4* floxed mouse to demonstrate a rescue of their phenotype (Figure 2 and extended Figure 2). *Smad7* is the iSmad that will affect all TGF β superfamily members and more precisely TGF β and BMP (unlike *Smad6* that is specific to the BMP pathway). The same applies to *Smad4* which is the co-Smad. *Smad4* forms a heterotetrameric complex with the R-Smads (*Smad2/3*-TGF β ; *Smad1/5/8*-BMP) then allowing translocation of the complex into the nucleus to affect target genes. As *Smad4* and *Smad7* are seen at the interface of both BMP and TGF β signaling pathways, the authors must demonstrate that the BMP signaling pathway is not affected in their model. Especially since BMP signaling plays such an important role in stem cell regulation in the GI tract and any deregulation in the BMP/WNT axis could have an impact by itself in some of the phenotype observed by the authors.

We thank the reviewer for this constructive suggestion and now provide additional data to address this query. Specifically, we have performed pair-wise gene set enrichment analyses (GSEA) alongside single-sample gene set enrichment analysis (ssGSEA) using the ssGSEA method in GSEA R package (v1.40.1) with murine TGF β and BMP gene sets retrieved from MsigDB. ssGSEA was applied to RNAseq data generated from intestinal tissue extracted from wild-type (*VilCre^{ER};Alk5^{+/+}*), *VilCre^{ER};Alk5^{CA}* (*Alk5^{CA}*), *VilCre;Smad4^{fl/fl}* (*Smad4^{KO}*) and *VilCre^{ER};Alk5^{CA};Smad4^{fl/fl}* (*Alk5^{CA}.Smad4^{KO}*) mice four days post tamoxifen-induction. Consistent with previous studies, we observe a positive enrichment of BMP signalling in wild-type intestinal tissue, which is not significantly altered following activation of TGFBR1 (*VilCre^{ER};Alk5^{CA}*) in the gut epithelium (Reviewer Fig. 1a). However compared to wild-type, deletion of *Smad4* (*VilCre;Smad4^{fl/fl}*) does drive a negative enrichment of BMP signature genes, which is in-line with its master regulatory function at the interface of BMP and TGF β signalling (Reviewer Fig. 1b). Importantly, activation of *Alk5* in concert with *Smad4* deletion

(*VilCre^{ER};Alk5^{CA};Smad4^{fl/fl}*) does not significantly change the enrichment of BMP signalling (Reviewer Fig. 1b; top heatmap plot), indicating upstream activation of TGFBR1 is insufficient to bypass the master regulatory function of Smad4 in controlling BMP signalling in the gut. We also looked at the expression of a select number of validated BMP target genes, including *Smad6*, across the different genotypes and observed a similar pattern of enrichment following loss of *Smad4*, but independent of Alk5 activation (*Alk5^{CA}*) (Reviewer Fig. 1b; bottom heatmap plot). To support our transcriptional observations, we stained intestinal tissue from wild-type (*VilCre^{ER};Alk5^{+/+}*), *VilCre^{ER};Alk5^{CA}*, *VilCre^{ER};Apc^{fl/fl}* and *VilCre^{ER};Apc^{fl/fl};Alk5^{CA}* mice four days post-tamoxifen for phospho-SMAD1/5 (pSMAD1/5) as an additional readout of BMP signalling (Reviewer Fig. 1c). The expression of pSMAD1/5 in wild-type mice was localised to villus epithelial cells and mostly absent in the crypt, which was comparable to *VilCre^{ER};Alk5^{CA}* mice (Reviewer Fig. 1c, see inset boxes for close-up of villus). This is consistent with the BMP gene set enrichment analysis (Reviewer Fig. 1a, b). Similarly, in the context of *Apc* mutation, Alk5 activation (*VilCre^{ER};Apc^{fl/fl};Alk5^{CA}*) does not induce noticeable changes to pSMAD1/5 expression in the villus compared to *VilCre^{ER};Apc^{fl/fl}* mice four days post-tamoxifen. Interestingly, we do observe expression of BMP antagonist *Grem1* localised to peri-cryptal mesenchymal cells, which is Smad4 dependent (Reviewer Fig. 1d). The basal expression of *Grem1* might go some way to explain the patterning of pSMAD1/5-positive epithelial cells along the crypt-villus axis in wild-type and *VilCre^{ER};Alk5^{CA}* epithelia. Collectively, these data indicate activation of epithelial TGF β signalling via *Alk5^{CA}* does not affect BMP signalling in the gut.

2. Figure 4b an b; In light of these results, the significant increase come from mostly 2 individuals from the 17 mutants analyzed compared to 12 controls. To this reviewer, this does not represent a clear shift in the phenotype as suggested by the authors. Thus, the biological significance of this results is overinterpreted and need to be readdressed accordingly.

We thank the reviewer for highlighting this discrepancy. We have reviewed and re-analysed the original tumour scoring data and an additional cohort of mice induced with a lower dose of tamoxifen (1mg). Both datasets comprise macroscopic and microscopic measurements of small and large intestinal tumours and upon reviewing the original tumour data, we detected an accidental error; tumour score duplication for one animal. As such, we have removed the duplication and re-plotted the tumour number and burden per mouse for the ageing cohorts shown in Fig. 4b and c. In agreement with the reviewer's observation, removal of the duplicated tumour scores shifts the statistical significance of the small intestinal tumour number and overall burden from P=0.041 to P=0.064 and P=0.01 to P=0.063 respectively. However, consistent with the original tumour dataset, *VilCre^{ER};Apc^{fl/+};Kras^{G12D/+};Alk5^{CA}* mice induced with 1mg tamoxifen had significantly accelerated tumour latency, but did not have significantly more small intestinal tumours than *VilCre^{ER};Apc^{fl/+};Kras^{G12D/+}* control mice (Reviewer Fig. 2a-c). As such, we have reworded the manuscript (lines 385-388) to indicate *VilCre^{ER};Apc^{fl/+};Kras^{G12D/+};Alk5^{CA}* mice had significantly accelerated tumour latency, which is consistent with the pro-tumourigenic role of TGF β signalling.

3. Reading of the manuscript is sometime very challenging and difficult to follow. I understand that the authors have numerous results to show and I commend them for that, but sometime the asynchronous sequence in the figures complexify the reading and breaks the flow for the reader. As for example there is a sequence in the paper where the reader has to go from Figure 4; extended Figure 5; extended Figure 6; Figure 6; extended Figure 6.

We thank the reviewer for this suggestion and agree that the manuscript was difficult to follow. We now have reordered figures in the manuscript to improve the flow (Figure 2 and the Extended Data Figs. 2-6) and significantly rewritten the manuscript to make it clearer.

Reviewer #2 (Remarks to the Author):

To date, the contribution of TGF β signaling to malignancy in CRC is controversial. Flanagan et al. use engineered mouse models and murine intestinal organoids to study the effect of constitutive active TGF β -signalling in the intestinal and colon epithelium (by conditionally expressing a mutant TGF β R1 that leads to ligand independent dimerization and activation of TGF β signaling). First, they show that active TGF β -signalling in the wildtype intestinal epithelium leads to apoptosis *in vitro* in organoids, but to enhanced cell proliferation *in vivo*. Second, they show that in an Apc mutant background constitutive TGF β activation leads to apoptosis in intestinal epithelial cells *in vitro* and *in vivo*. Third, they found that if in addition to the APC mutation intestinal epithelial cells also contain a Kras-G12D mutation they are protected from TGF β mediated apoptosis. In addition, they demonstrate that in this background TGF β expression leads to increased malignancy. Interestingly, they also found that the gene expression signature of these mice correlates with the signature of a highly aggressive T1 subtype of CRC in patients, and they demonstrate that inhibition of MEK and EGFR counteracts the effect of overactive TGF β -signalling in tApc-KrasG12D mice.

I recommend this manuscript for publication in Nature Communications. The experimental design of this study is coherent and the work adds substantial evidence to the controversy of TGF β -signaling in CRC. In addition, it is a valuable information that MEK inhibition could be beneficial in tumors with overactive TGF β signaling. However, resolving the following comments may add to the quality of the manuscript.

Main points

- The work heavily relies on findings derived by autochthonous mouse models and corresponding murine organoid lines. Proofing key aspects of the proposed mechanisms in human cells would allow to generalize the conclusions of this study to human cancer. This may be achieved by testing TGF β response in human intestinal organoids that contain LOF mutations in APC (using CRISPR) and express KrasG12D (using a lentiviral vectors from addgene).

We thank the reviewer for this suggestion and understand that while our current manuscript relies heavily on genetically-engineered-mouse-models to study fundamental cancer biology *in vivo*, there is scope for demonstrating how well these findings translate using human systems. As suggested by the reviewer, we have utilised CRISPR-edited human colonic organoids with defined APC and KRAS mutations (Drost et al. 2015 Nature) to study the effects of TGF β *in vitro*.

Similar to our experimental design using murine-derived organoid models, APC.KRAS-mutant organoids were seeded as single cells and treated with vehicle or recombinant TGF β 1 48hrs after plating. Cell viability was measured 5 days after treatment (Reviewer Fig. 3a, b; P0 data). After this timepoint, organoids were mechanically passaged and re-seeded as single-cells and re-treated 48hrs after plating. Cell viability was recorded 5 days after treatment (Reviewer Fig. 3a, b; P1 data). In contrast to our observations in murine-derived Apc.Kras-mutant intestinal organoids (Extended Data Fig. 5a-c in the revised manuscript), human APC.KRAS-mutant colonic organoids treated with TGF β 1 showed significant growth and viability impairment (Reviewer Fig. 3a, b). The observed TGF β 1-mediated growth and viability deficits were exacerbated following mechanical organoid passage, which was also observed in murine-derived organoids (Extended Data Fig. 5a-c in the revised manuscript). However, despite the different sensitivities in response to TGF β 1, our data fit with previous studies that show TGF β is effective at killing individual/isolated human cells with Wnt- and MAPK-activating mutations (Matano et al. 2015 Nat. Med.). This could be attributed to different growth-factor requirements between mouse and human organoids, and given the absence of a surrounding stroma in organoid models, it is tempting to speculate that additional stromal-derived niche signals are required to support epithelial cells against the cytostatic effects of TGF β . It interesting to note that *VilCre;Apc^{fl/fl};Kras^{G12D/+};Alk5^{CA}* mouse organoids retained expression of the ALK5

transgene (*Alk5^{CA}*) and grew well (Reviewer Fig. 3c and Extended Data Fig. 5d), suggestive that there is a “just right” threshold for promoting a level of TGFβ in the organoids, which is in contrast to *VilCre;Apc^{fl/fl};Alk5^{CA}* organoids. Furthermore, the signatures derived from our *VilCre;Apc^{fl/fl};Kras^{G12D/+};Alk5^{CA}* mouse model significantly overlap with human tissue and can be used in unison to identify epithelial TGFβ is a predictor of aggressive human disease (much better than the GSEA enrichment signature for TGFβ, most of which have been derived from cell lines). Therefore, whilst we note (and thanks to the reviewer for their comments and suggested experiments) there are species-specific differences in organoids, we believe that together with the murine and human *in vivo* data we have elucidated an important conserved role for epithelial TGFβ in driving a poor prognosis in CRC.

- For some of the bar graphs that quantify histology results no example images are shown (for example Figure 2e,f). Here it would be interesting to see in which regions of the epithelium cells are apoptotic or proliferating.

We thank the reviewer for this suggestion and have included representative images of BrdU incorporation and apoptosis in *VilCre^{ER};Alk5^{+/+}* and *VilCre^{ER};Alk5^{CA}* tissue post-tamoxifen induction in the revised manuscript (Fig. 2d). In contrast to wild-type gut, which have stereotypical crypt-based BrdU⁺ cells and apoptotic cells present in the villi, *VilCre^{ER};Alk5^{CA}* animals have apoptotic cells present in the crypt, which may partially explain the enlarged crypt phenotype and increased cell proliferation and number of *Olfm4⁺* intestinal stem cells following constitutive *Alk5^{CA}* expression (Fig. 2f, g).

- Figure 2: It is somehow surprising that in this experimental setup constitutive active TGFβ did not perturb the stem cell niche. Were mice also followed up mice for longer periods than 1 week after Tmx treatment?

We thank the reviewer for this insightful observation and provide additional *in vivo* data to address this question. To examine potential changes to the stem cell niche following constitutive *Alk5^{CA}* expression, we originally documented changes in Paneth/lysozyme⁺ cell number and location in the gut epithelium as well as increases of *Olfm4⁺* crypt cells (Fig. 2d, g, h). However, to assess potential longer-term impact on the stem cell niche, we aged *VilCre^{ER};Alk5^{CA}* animals for 60 days post-tamoxifen induction. Interestingly, the disruption to Paneth and *Olfm4⁺* cells observed four days post-tamoxifen (Fig. 2d, g, h) was resolved at 60 days post-tamoxifen. This new data is included in the revised manuscript as Extended Data Fig. 2c. Of note, this result is not entirely surprising as the intestinal epithelium has well-documented capabilities of adapting to various physiologic, genetic and mechanical challenges (Beumer & Clevers 2016, Development).

- Figure 3: Was the effect of TGF-beta activation also assessed in other Wnt activating backgrounds such as Axin2 knock out? This could be done by genetically disrupting Axin2 using CRISPR in murine intestinal VilCreER x AlkCA organoids, and testing whether this line behaves similar as Apc-mutant VilCreER x AlkCA or organoids.

We thank the reviewer for this intriguing and important question. To address this question, we modelled Wnt hyperactivation by genetically targeting Exon 3 of *CTNNB1* (β-catenin; β-cat^{Ex3}), which encodes serine and threonine sites phosphorylated by GSK3-β to promote β-catenin degradation. We chose this approach in preference to genetic disruption of Axin2 gene function because Axin1 is considered to be functionally equivalent to Axin2 *in vivo* (Chia & Costantini 2005 Mol. Cell. Biol., Feng et al. 2012 Gastroenterology), where it can compensate and regulate the degradation of β-catenin in the absence of Axin1. As such, we generated small intestinal organoid cultures from *VilCre^{ER};β-cat^{Ex3/Ex3}* mice four days post-tamoxifen injection and treated the cultures with recombinant TGFβ1 over the course of four days to mimic constitutive TGFβ activation, akin to *VilCre^{ER};Apc^{fl/fl};Alk5^{CA}*. Consistent with the

phenotype observed in *Apc*-mutant organoid cultures subjected to constitutive TGF β activation, *VilCre^{ER}; β -cat^{Ex3/Ex3}* cultures treated with TGF β 1 underwent rapid cell death and reduced viability. We have included this new data in Extended Data Fig. 3c and lines 323-331 of the revised manuscript.

Additionally, we examined gut homeostasis in *VilCre^{ER}; β -cat^{Ex3/+}* and *VilCre^{ER}; β -cat^{Ex3/+};*Alk5^{CA}* mice four days post-tamoxifen induction. While *VilCre^{ER}; β -cat^{Ex3/+}* and *VilCre^{ER}; β -cat^{Ex3/+};*Alk5^{CA}* mice show comparable crypt hyperplasia (Reviewer Fig. 4a, see H&E staining), pSMAD3 expression is lower in *VilCre^{ER}; β -cat^{Ex3/+}* crypt epithelial cells compared to *VilCre^{ER}; β -cat^{Ex3/+};*Alk5^{CA}* mice (Reviewer Fig. 4a, see pSMAD3 panels), indicating differential TGF β activation via *Alk5^{CA}* in the context of *β -cat^{Ex3/+}* mutation. This finding might be attributed to the slower kinetics of *β -cat^{Ex3/+}*- vs *Apc^{fl/fl}*-mutant cells to reach optimal activation of the Wnt pathway, which we have previously described (Huels et al. 2015 EMBO J.), and thus allow *β -cat^{Ex3/+}* epithelial cells to better tolerate epithelial *Alk5^{CA}* expression compared to *Apc^{fl/fl}* cells at this timepoint (four days post-tamoxifen). Indeed, the frequency of epithelial cells with 'hyperactivated Wnt', as determined by nuclear β -catenin IHC, is lower in *VilCre^{ER}; β -cat^{Ex3/+}* and *VilCre^{ER}; β -cat^{Ex3/+};*Alk5^{CA}* mice compared to *VilCre^{ER};*Apc^{fl/fl}** four days post-tamoxifen (Reviewer Fig. 4b). In support, *VilCre^{ER}; β -cat^{Ex3/+};*Alk5^{CA}* mice partially retain expression of *Alk5^{CA}* transgene, suggesting epithelial *Alk5^{CA}* expression is better tolerated in *β -cat^{Ex3/+}*-mutant cells compared to *Apc*-mutant cells due to the relative lower activation of Wnt signalling (Reviewer Fig. 4c and Extended Data Fig. 3e). To further support our hypothesis, we examined the intestines of *VilCre^{ER}; β -cat^{Ex3/+}* and *VilCre^{ER}; β -cat^{Ex3/+};*Alk5^{CA}* mice 20 days post-tamoxifen and observed robust Wnt activation (nuclear β -catenin) in *VilCre^{ER}; β -cat^{Ex3/+}* and *VilCre^{ER}; β -cat^{Ex3/+};*Alk5^{CA}* mice, confirming the relative slower kinetics of *β -cat^{Ex3/+}* mutation to transform the gut (Day 4 vs Day 20). Critically, at this time point, *VilCre^{ER}; β -cat^{Ex3/+};*Alk5^{CA}* epithelial cells were mostly devoid of *Alk5^{CA}* expression and had reduced pSMAD3 expression in crypt cells, which further support the premise that cells with hyperactivated Wnt are highly sensitive to epithelial TGF β activation (Reviewer Fig. 4d).********

Finally, we generated *VilCre^{ER}; β -cat^{Ex3/+};*Kras^{G12D/+};*Alk5^{CA}* mice to determine whether *Kras* mutation in concert with *β -cat^{Ex3/+}* confers tolerance to epithelial TGF β activation. Consistent with our findings in *Apc*-mutant mice (Fig. 5 and Extended Data Fig. 6 in the revised manuscript), *VilCre^{ER}; β -cat^{Ex3/+};*Kras^{G12D/+};*Alk5^{CA}* mice retain *Alk5^{CA}* transgene expression (Reviewer Fig. 4c, right-most panel), supporting our hypothesis that activation of MAPK helps epithelial cells with active Wnt to withstand epithelial TGF β activation. Taken together, these new data suggest and support that Wnt-activating mutations, such as *Apc* or *CTNNB1*, sensitise the gut epithelium to TGF β activation and expose a need for tumour cells to acquire additional mutations, such as *Kras*, that can tolerate constitutive/chronic TGF β activation.****

- Extended Fig 6a: How often were organoids passaged in TGF β medium? In the brightfield image some organoids appear apoptotic. It would be important to show viability after passaging. In addition, it seems as in the study of Drost et al. human small intestinal organoids mutant for *Kras^{G12D}* and *Apc* cannot grow without Noggin (see publication Drost et al. Nature 2017). Thus, it would be interesting to also perform experiments in human organoids in this study (see point 1). If the outcome in human organoids is not identical to murine organoids I would still recommend publication of the manuscript, but this information would be valuable for the field.

We thank the reviewer for this important question and now provide additional data to clarify the effect of TGF β signalling in the context of *Apc* and *Kras^{G12D}* mutation *in vitro*. In the original submission, the data presented in Extended Data Fig. 6a-c (now Extended Data Fig. 5a-c in the revised manuscript) is of *VilCre^{ER};*Apc^{fl/fl};*Kras^{G12D}** organoids treated with recombinant TGF β 1 for a duration of five days, after which point cell viability and the expression of several TGF β -target and apoptotic genes were tested. During the course of the experiment, the organoids were not passaged, as the experiment lasted only five days. We have repeated*

these experiments, but this time quantifying organoid viability after passage as suggested. In contrast to our original observations (Extended Data Fig. 6a-c), the viability of TGFβ1-treated *VilCre^{ER};Apc^{fl/fl};Kras^{G12D}* organoids following passage is significantly reduced compared to vehicle-treated *VilCre^{ER};Apc^{fl/fl};Kras^{G12D}* organoids (Extended Data Fig. 5b; P1 data in the revised manuscript). We have referenced this new finding in lines 401-407 of the revised manuscript. This result is not entirely unexpected as the effect of genetic and/or pharmacological perturbations after serial passage of 3D cultures is often heightened due to the requirement of cells to undergo a regenerative-like response (Flanagan et al. 2015 Stem Cell Reports). This can be in contrast to the effects seen in established organoid cultures, like those presented in Extended Data Fig. 6a-c of the original manuscript. Of note, our original findings are consistent with previous studies documenting how mutant *Kras* models can tolerate TGFβ1 treatment for several days *in vitro* (Wiener et al. 2014 PNAS).

Finally, to determine whether the same cytostatic effects of TGFβ1 in murine organoid cultures translate to human organoid cultures we performed the experiments outlined above – please see answer to point #1.

Minor points

- Do the authors have a hypothesis what could protect the intestinal epithelium *in vivo* of strong cell death induction compared to epithelial-only organoid cultures? Are there factors secreted from the stroma/mesenchyme that could mediate this effect?

We thank the reviewer for this thought-provoking question and are excited to report we have preliminary data to suggest the BMP antagonist Gremlin1 (*Grem1*) is upregulated by sub-epithelial stromal cells in *VilCre^{ER};Alk5^{CA}* animals (please see Reviewer Fig. 1d). Upregulation of *Grem1* by the surrounding stroma may help explain the notable difference in phenotypes (*in vitro* vs *in vivo*) and the significant increase in epithelial cell proliferation and intestinal stem cell number in response to constitutive TGFβ activation *in vivo* (Fig. 2d, f, g). This hypothesis is entirely consistent with previously reported functions for *Grem1* in both the epithelium and stroma (Davis et al., 2015 Nat. Med., McCarthy et al., 2020 Cell Stem Cell). The functional significance of *Grem1* is under investigation and will form the basis for future publication.

- Lines 158-160: A statistical analysis would help to interpret the data, even though I acknowledge that the sample size is relatively small and the derived p-values should be interpreted with caution.

To account for small sample sizes, we utilised the CRIS and CMS classifiers in conjunction with GSEA, where we observed uniform elevation of CRIS signalling in all relapse cases compared to non-relapse, suggesting that aggressive pT1 tumours have activated multiple intrinsic cancer signalling pathways (in the revised manuscript Extended Data Fig. 1d, padj=5.6e-09; 0.3; 0.1; 0.12 and 0.003 respectively). The same analyses for CMS revealed that, in line with our stromal assessments (Fig. 1a), aggressive pT1 tumours had significantly higher levels of signalling associated with epithelial biology (CMS2, padj=0.01 and CMS3, padj=2.6e-09) and a significant reduction in stromal signalling (CMS1, padj=0.003 and CMS4, padj=3.7e-10), particularly the fibroblast-related subtype CMS4. The p-values listed above have been incorporated in figure legend of Extended Data Fig. 1d in the revised manuscript.

Reviewer #3 (Remarks to the Author):

This manuscript describes exciting research to characterise the role of TGFβ in “born-to-be bad” early-stage colorectal cancers (CRC). Interestingly, TGFβ is associated with stroma in late-stage CRC, whereas the same TGFβ is related to the epithelium of the early-stage CRC. The study includes a range of *in vivo* models that addresses the role of TGFβ in the presence of APC/KRAS aberrations. Following are a few comments:

- Extended Fig1 shows PCA with separation of relapse and non-relapse samples by PC2. It would be good to show the clustering by k-means or other methods within the PCA plot.

We thank the reviewer for a very interesting point. Potentially a combination of PCA and Kmeans algorithm is a suitable way to analyse this data when dealing with large datasets to see if the revealed clusters would be associated with the existing phenotype. However, our novel T1 data is a small patient cohort (n=27), and we were interested to find molecular signatures discriminating the relapse cases from non-relapse. Our aim wasn't to provide a refined gene classifier, given the sample size and potential for over-fitting, but to identify differential biological signalling in general that distinguishes aggressive early-stage tumours. As suggested by the reviewer, we have now performed kmeans clustering on the dataset and replotted the PCA with the corresponding groups identified (please see Reviewer Fig. 5a). Interestingly, both analyses show the same trend to some extent, with the kmeans clusters predominantly aligning with PC1 of the PCA analysis, despite PCA not being a clustering method.

- It is unclear what the significance of Figures 1a and b. Shouldn't these be supplementary figures?

We thank the reviewer for this suggestion, but respectfully disagree as our intention/motivation for including data shown in Figs 1a and b is to highlight that relapse status in our T1 cohort is not associated with previously-defined prognostic histological features (fibroblasts or cytotoxic lymphocytes). Therefore, these data assist the narrative of the manuscript towards investigating epithelial-centric biology. We feel presenting this data from the outset as a main figure is very important to **a)** recognise and address the well-established role of immune and stromal cell types in promoting aggressive CRC and **b)** aids the flow of the manuscript and the consequent ability of the reader to understand the logic and rationale behind the ensuing experiments and analysis.

- It would be helpful to include Ext. Fig 1C as main figures.

We thank the reviewer for this suggestion and now present CRIS and CMS subtype positioning of relapse vs non-relapse patients as normalised enrichment scores in Fig. 1e of the revised manuscript. We have also kept individual GSEA plots for each CRIS and CMS subtype in Extended Data Fig. 1d in the revised manuscript.

- "CMS4 and CRIS-B have previously been associated with poor prognosis in later stage disease". The authors need to check this statement for validity and cite them accordingly.

We thank the reviewer for highlighting this textual oversight. We have now included relevant references in the revised manuscript (line 212) to support the claims that CMS4 and CRIS-B subtypes of CRC are associated with poor prognosis in advanced stage CRC.

- References to certain statements from literature are missing.

We thank the reviewer for highlighting this textual oversight. We have reviewed the manuscript and included additional references that were absent in the original submission. Specifically lines 108, 120, 219 and 310 in the revised manuscript

- Where comments about correlation have been made (significantly higher or lower), P values and statistical tests applied are not always stated.

We thank the reviewer for this comment and have listed the P values and statistical tests in the figure and figure legend where correlation has been described. Specifically, these analyses revealed a significant positive correlation between transcriptome-based stromal

(fibroblast) scores and the digital pathology-based stromal scores (Fig. 1b, $\rho=0.69$, $p=0.00009$), validating the in-silico approach. We and others have previously reported a significant association between TGF β signalling and the levels of both tumour infiltrating fibroblasts and cytotoxic immune cells in later stage disease, however this association was not observed in our pT1 tumours (Fig. 1g; $\rho=-0.1$, $p=0.6$ and Extended Data Fig. 1e; $r=0.58$, $p<2.2e-16$ in the revised manuscript).

- It would be great to highlight the potential translational aspect of the findings.

We thank the reviewer for this suggestion and now provide additional text in the discussion (lines 679-696 in the revised manuscript) to elevate translational aspect of our study in the context of targeted therapies for *KRAS*- and *BRAF*-mutant CRC.

Reviewer Fig. 1 Constitutive TGFβ/ALK5 activation does not perturb BMP signalling in the gut

a, Group-wise GSEA was performed using fgsea R package on ranked genes based on shrunken log expression ratio. NES, normalized enrichment score. padj (adjusted p-value), FDR-corrected p-values. **b**, Top, single-sample gene set enrichment analysis (ssGSEA) was performed, using the ssGSEA method in GSVA R package with the murine TGFβ and BMP gene sets retrieved from MsigDB. Heatmap were generated in R by first scaling the data using the 'scale' function from the base package to generate Z-scores, then plotting them using the 'pheatmap' function from pheatmap package. Red in the colour bar indicates positive enrichment and blue shows negative enrichment. Bottom, heatmap of scaled normalized expression value across mice for BMP target genes. **c**, Representative phospho-SMAD1/5 (pSMAD1/5) staining on intestinal tissue from wild-type (*VilCre^{ER};Alk5^{+/+}*), *Alk5^{CA}* (*VilCre^{ER};Alk5^{CA}*), *Apc* (*VilCre^{ER};Apc^{fl/fl}*) and *Apc Alk5^{CA}* (*VilCre^{ER};Apc^{fl/fl};Alk5^{CA}*) mice four days post-tamoxifen induction. Inset boxes and arrows indicate pSMAD1/5 positive cells on the villus, but not crypt. Scale bar, 100 μm. **d**, Representative staining of Gremlin1 (*Grem1*) on intestinal tissue from wild-type (*VilCre^{ER};Alk5^{+/+}*), *Alk5^{CA}* (*VilCre^{ER};Alk5^{CA}*) and *Alk5^{CA} Smad4^{KO}* (*VilCre^{ER};Alk5^{CA};Smad4^{fl/fl}*) mice four days post-tamoxifen induction. Inset boxes show *Grem1* positive cells are localised around the crypt base. Scale bar, 100 μm

Reviewer Fig. 2 Epithelial TGF β /ALK5 signalling synergises with Wnt and MAPK signalling to accelerate tumorigenesis.

a, Survival plot for *VilCre^{ER};Apc^{fl/+};Kras^{G12D/+};Alk5^{+/+}* (n=10) and *VilCre^{ER};Apc^{fl/+};Kras^{G12D/+};Alk5^{CA}* (n=15) mice aged until clinical endpoint following 1mg tamoxifen induction. $P=1.0 \times 10^{-4}$, log-rank test. **b-c**, Small intestinal (SI) (**b**) and large intestinal (LI) (**c**) tumour number per mouse scored off H&E stained sections from mice described in **a**. Each data point represents individual mice. Data are \pm s.e.m; $P=0.348$ (**b**), **** $P < 0.0001$ (**c**). Mann-Whitney U-test.

Reviewer Fig. 3 Human intestinal organoids with concurrent *APC* and *KRAS* mutation are sensitive to TGFβ stimulation.

a, Representative images of *APC.KRAS*-mutant human colonic organoids, generated using CRISPR/Cas9 editing, treated with vehicle (PBS/BSA) or TGFβ1 (5 ng/ml) after plating (P0) and serial passage (P1). Briefly, ~30K single-cells per well were plated on d0 and treated with vehicle (PBS/BSA) or TGFβ1 (5 ng/ml) 2 days after plating. Treated P0 organoids were mechanically passaged and re-plated as single cells and re-treated with vehicle or TGFβ1 2 days after being plated (P1 panels). Images were taken 5 days after initial seeding (P0) and 5 days after serial passaging (P1). Scale bar, 100 μm. **b**, Relative viability of human organoids described in **a**. Cell viability was measured 5 days following initial seeding (P0), and 5 days after serial passage (P1). Individual data points indicate independent mutant-organoid lines; *P=0.02. All data are ± s.e.m; Mann–Whitney U-test. **c**, qPCR for ALK5 transgene (*Alk5^{CA}*) in *VilCre^{ER};Apc^{fl/fl};Kras^{G12D/+};Alk5^{CA}* organoids following in vitro recombination (Tmx). n=3 independent organoid lines. *P=0.05. All data are ± s.e.m; Mann–Whitney U-test.

Reviewer Fig. 4 Activation of WNT and MAPK together, but not WNT alone tolerises *Alk5^{CA}* activation.

a, Representative H&E and pSMAD3 staining on tissue from mice of the indicated genotypes four days post-tamoxifen induction. Scale bar, 200 μ m. Note, *Bcat^{Ex3/+}; Alk5^{CA}* epithelial cells minimally express *Alk5^{CA}* transgene. Please note, *Alk5^{CA}* staining for *Bcat^{Ex3/+}* and *Bcat^{Ex3/+}; Kras^{G12D/+}* not shown as these animals do not express *Alk5^{CA}* transgene. Scale bar, 200 μ m. **b**, Representative β -catenin IHC on tissue from mice of the indicated genotypes four days post-tamoxifen induction. Scale bar, 200 μ m. **c**, Representative *Alk5^{CA}* ISH staining on tissue from mice of the indicated genotypes four days post-tamoxifen induction. Please note, *Alk5^{CA}* staining for *Bcat^{Ex3/+}* and *Bcat^{Ex3/+}; Kras^{G12D/+}* not shown as these animals do not express *Alk5^{CA}* transgene. Scale bar, 200 μ m. **d**, Representative β -catenin and pSMAD3 IHC and *Alk5^{CA}* ISH staining on tissue from mice of the indicated genotypes twenty days post-tamoxifen induction. Please note, *Alk5^{CA}* staining for *Bcat^{Ex3/+}* not shown as these animals do not express *Alk5^{CA}* transgene. Scale bar, 200 μ m.

Reviewer Fig. 5 PCA was performed using prcomp function in stats R package, then K-means method applied to the PCA scores in order to form clusters. Two kmeans clusters are indicated by circle and triangle.

REVIEWERS' COMMENTS

Reviewer #1 (Remarks to the Author):

The authors have addressed all my concerns in this revised version of their manuscript.

This manuscript is suitable for publications in Nature Communications.

Reviewer #2 (Remarks to the Author):

The authors have addressed all my concerns and I can recommend publication of the article at Nat. Communications.

Reviewer #3 (Remarks to the Author):

The authors have addressed all the comments.

2nd revision:

Reviewer #1 (Remarks to the Author):

The authors have addressed all my concerns in this revised version of their manuscript.

This manuscript is suitable for publications in Nature Communications.

We thank the reviewer for their support and helpful comments throughout the review process.

Reviewer #2 (Remarks to the Author):

The authors have addressed all my concerns and I can recommend publication of the article at Nat. Communications.

We thank the reviewer for their support and helpful comments throughout the review process.

Reviewer #3 (Remarks to the Author):

The authors have addressed all the comments.

We thank the reviewer for their support and helpful comments throughout the review process.